# Advancing Cross-domain Discriminability in Continual Learning of Vision-Language Models

**Yicheng Xu**[1*]  **Yuxin Chen**[2*]  **Jiahao Nie**[3]
**Yusong Wang**[1]  **Huiping Zhuang**[4,5†]  **Manabu Okumura**[1]

[1]Institute of Science Tokyo  [2]University of California Berkeley
[3]Nanyang Technological University  [4]South China University of Technology
[5]Greater Bay Area Institute for Innovation, Hunan University, China

`yxu040@e.ntu.edu.sg, yuxinc@berkeley.edu`

## Abstract

Continual learning (CL) with Vision-Language Models (VLMs) has overcome the constraints of traditional CL, which only focuses on previously encountered classes. During the CL of VLMs, we need not only to prevent the catastrophic forgetting on incrementally learned knowledge but also to preserve the zero-shot ability of VLMs. However, existing methods require additional reference datasets to maintain such zero-shot ability and rely on domain-identity hints to classify images across different domains. In this study, we propose **R**egression-based **A**nalytic **I**ncremental **L**earning (RAIL), which utilizes a recursive ridge regression-based adapter to learn from a sequence of domains in a non-forgetting manner and decouple the cross-domain correlations by projecting features to a higher-dimensional space. Cooperating with a training-free fusion module, RAIL absolutely preserves the VLM's zero-shot ability on unseen domains without any reference data. Additionally, we introduce **Cross**-domain **T**ask-**A**gnostic **I**ncremental **L**earning (X-TAIL) setting. In this setting, a CL learner is required to incrementally learn from multiple domains and classify test images from both seen and unseen domains without any domain-identity hint. We theoretically prove RAIL's absolute memorization on incrementally learned domains. Experiment results affirm RAIL's state-of-the-art performance in both X-TAIL and existing Multi-domain Task-Incremental Learning settings. The code is released at `https://github.com/linghan1997/Regression-based-Analytic-Incremental-Learning`.

## 1 Introduction

Continual learning (CL) [1, 2, 3] is a crucial area in machine learning, which requires a learner to incrementally learn new data instead of training from scratch. The main challenge in CL is known as catastrophic forgetting [4], where learning new knowledge results in the forgetting of the old one. To this end, various CL approaches [5, 6, 7, 8, 3, 9] have been proposed to solve the forgetting issue. As a typical CL setting, Class-Incremental Learning (CIL) (Fig. 1 (a)) aims to achieve robust discriminability on all seen classes. Despite the advancements, existing approaches mainly focus on classifying images only from seen classes, thereby limiting the model's generalizability.

Consequently, Zheng *et al.* [10] proposed Multi-domain Task-Incremental Learning (MTIL), which cooperates CL with the zero-shot ability of Vision-Language Models (VLMs) [11, 12, 13, 14] such as CLIP [13]. This integration equips models with the ability to classify domains they have not

---

[*]Equal contribution
[†]Corresponding author: hpzhuang@scut.edu.cn

38th Conference on Neural Information Processing Systems (NeurIPS 2024).

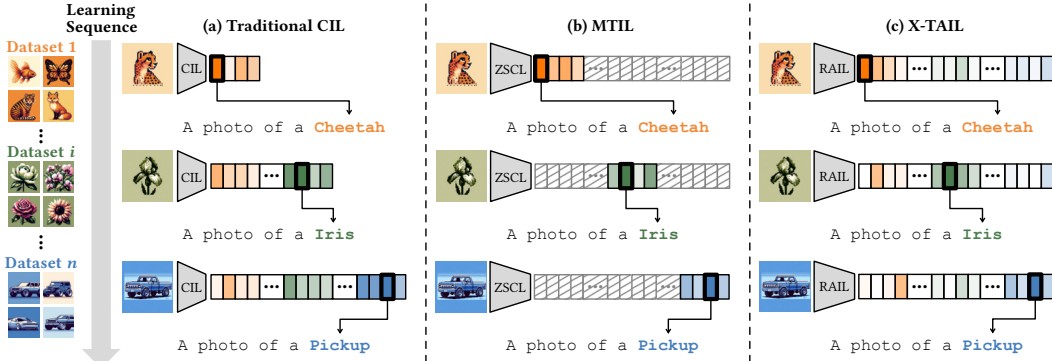

Figure 1: **Comparison of different CL settings.** (a) In CIL, models classify images within all previously encountered classes. (b) In MTIL, models classify images from both seen and unseen domains based on the given domain-identities. (c) In X-TAIL, models classify images from both seen and unseen domains without any domain-identity hint.

yet encountered, enhancing their generalizability across multiple domains (Fig. 1 (b)). Several methods [10, 15] have been specifically designed for MTIL, in which the model is required to retain both the incrementally learned knowledge during CL and the zero-shot ability of VLMs. However, these methods require a domain-identity hint to indicate the specific domain of the test image, which is often not applicable in real-world scenarios [7]. Additionally, the use of reference datasets during training is necessary to maintain pre-trained VLMs' zero-shot performance.

To address the aforementioned limitations, we introduce **R**egression-based **A**nalytic **I**ncremental **L**earning (RAIL), a novel approach that incrementally learns new knowledge and performs effectively on both seen and unseen domains. Specifically, we leverage non-linear projection functions from both primal and dual perspectives to enhance the expressiveness of features extracted by the pre-trained CLIP. It endows the learner with the ability to classify images in a cross-domain label set without any domain-identity hint. In the incremental learning process, RAIL utilizes a ridge regression-based adapter that updates the parameters recursively. This is identical to learning on all encountered domains at once, achieving absolute memorization on learned domains. Additionally, we freeze the pre-trained CLIP and design a training-free fusion module to determine whether the test data belongs to seen or unseen domains. This strategy absolutely preserves CLIP's zero-shot ability on unseen domains, meeting practical requirements for models deployed in dynamic environments.

To demonstrate the effectiveness of our method, we propose **Cross**-domain **T**ask-**A**gnostic **I**ncremental **L**earning (X-TAIL) setting as illustrated in Fig. 1 (c). Particularly, X-TAIL requires CL methods to incrementally transfer a pre-trained VLM to multiple domains while evaluating the model's performance on both seen and unseen domains. Moreover, domain hints are forbidden in X-TAIL, making it more realistic and challenging [3]. As a result, effective CL methods must classify a test image into the correct domain and class simultaneously.

Our contributions are summarized as follows:

- We propose a new CL method RAIL to incrementally adapt the pre-trained VLM to multiple domains without forgetting both pre-trained and incrementally learned knowledge.

- To meet the practical scenario where CL methods need to sequentially learn data from new domains and classify images across these domains, we propose a new setting X-TAIL to evaluate the preservation of VLM's zero-shot ability and the adaptability to new domains.

- We theoretically prove the RAIL's absolute memorization on incrementally learned domains and demonstrate that the zero-shot ability of the pre-trained VLM on unseen domains is absolutely preserved.

- We empirically show that the proposed method achieves state-of-the-art performances on both existing MTIL and the novel X-TAIL settings.

## 2 Related work

Early CL methods focused on Task-Incremental Learning (TIL) [16], where a task-id is given during testing. Subsequently, a more practical and challenging setting of Class-Incremental Learning (CIL) [7] was proposed, where the access to the task-id is forbidden at inference time. Methods for CIL must therefore distinguish between all classes encountered in learned tasks. More recently, Zheng *et al.* [10] proposed Multi-Domain Task-Incremental Learning (MTIL), which is especially designed to evaluate CL methods with pre-trained VLMs. In MTIL, a pre-trained VLM continually adapts to multi-domain tasks. The performance on both seen and unseen tasks measure the retention of both incrementally acquired and pre-trained knowledge. However, it still requires the task-id to create specific domain label space at inference time. Apart from them, X-TAIL combines the challenges of both CIL and MTIL, in which the model learns new classes from various incoming domains and distinguishes between both seen and unseen classes without any domain-identity.

Prevailing continual learning methods include replay-based, distillation-based, regularization-based, and architecture-based approaches [3]. Replay-based methods such as iCaRL [7] typically store a small portion of the previous task data as exemplars. The model is then trained jointly on new task data and the saved exemplars to preserve the previous knowledge. Distillation-based methods such as LwF [6] use either weight or function regularization to transfer knowledge from the previous model to the current model for knowledge distillation. Regularization-based methods such as ZSCL [10] penalize the shift of either model parameter or feature space by adding a regularization term to the cross-entropy loss function. To preserve the robustness of the strong pre-trained model without access to the pre-trained dataset, ZSCL utilizes large-scale reference datasets to regularize the parameter space. Architecture-based methods [17, 18, 19] expand the model by constructing task-specific parameters to avoid inter-task interference. For example, MoE-Adapters [15] cooperates the pre-trained CLIP with mixture of experts (MoE) [20] to learn from different domains. By leveraging a reference dataset to initialize a task-id indicator, it enables the model to distinguish unseen tasks from seen ones.

The aforementioned methods either neglect the forgetting issue of pre-trained knowledge or require multiple iterations and large-scale reference datasets for training, making it challenging to efficiently adapt to new data in continual learning scenarios. By contrast, RAIL employs an analytical solution that achieves the optimum in a single epoch without additional reference data, ensuring its efficiency.

## 3 Cross-domain task-agnostic incremental learning

### 3.1 Problem setting

We define Cross-domain Task-Agnostic Incremental Learning (X-TAIL) as follows. Given a pre-trained VLM, the learner is required to incrementally transfer it to $N$ different domains $\{D^{(1)}, D^{(2)}, ..., D^{(N)}\}$. Each domain $D^{(n)} = \{(\mathbf{x}_j^{(n)}, y_j^{(n)})\}_{j=1}^{|D^{(n)}|}$ is available only during the $n$-th learning step. The class labels $y_j^{(n)} \in C_{\text{label}}^{(n)}$ from the incrementally learned domain $D^{(n)}$ are added to the set of seen class labels. During inference at all steps, the learner attempts to classify input images from any domain without the domain-identity hint. In other words, the ground-truth label of the test image belongs to $C_N = C_L \cup C_U$, where $C_L = \bigcup_{i=0}^{n} C_{\text{label}}^{(i)}$ is the union of seen class labels from all previous learning steps and $C_U$ is the set of unseen class labels.

Similar to the task-id in TIL, the domain-identity hint allows the learner to classify input data within the label space of a specific domain during evaluation. Essentially, the learner knows the domain of test images, which is far from real-world application scenarios. For instance, the learner is supposed to predict an image as the class of *husky* from all possible classes $C_N =$*{car, bus, ..., churros, donuts, ..., husky, beagle, bulldog, ...}*. However, if the domain-identity hint is given, the learner only needs to predict from a limited subset $C_{\text{dog}} =$*{husky, beagle, bulldog, ...}*, which is simpler but less realistic compared to practical applications. Therefore, we extend our setting to a task-agnostic scenario. Specifically, the learner predicts images from $C_N$, the union of any potential class labels, without any domain hint.

## 3.2  Datasets

In X-TAIL's cross-domain setting, the learner should encompass as extensive data distributions as possible. Following previous work [10, 15, 21, 13, 22, 23], we select 10 different image-classification datasets from different domains for our setting: Aircraft [24], Caltech101 [25], DTD [26], EuroSAT [27], Flowers [28], Food [29], MNIST [30], OxfordPet [31], StanfordCars [32], and SUN397 [33]. Specifically, to prevent the redundancy of learning overlapping classes and to maintain the integrity of the setting, CIFAR-100 [34] was excluded because it includes many classes that overlap with those in other domains. As a result, CL methods under X-TAIL should discriminate images from a total of 1, 100 classes across all domains.

## 3.3  Evaluation metrics

We adopt the evaluation metrics from [10] for our setting. As illustrated in Fig. 2, each column represents the performance on a specific domain after each learning step, while the rows correspond to the learning sequence. In traditional CL settings, only the results in the lower diagonal where the learner has been exposed to the exemplars from the test domains are measured. Nevertheless, X-TAIL extends this evaluation to cover the entire matrix, recording performances across both learned and unlearned domains. The *"Average"* metric, averaged on the orange blocks, indicates the average accuracy of all learning steps across all domains. The gray and green blocks under the diagonal show the classification performance on these domains after the model has learned these domains. Specifically, the green blocks represent the model's last performance on these domains after learning all domains. The *"Last"* metric, which is the average of the

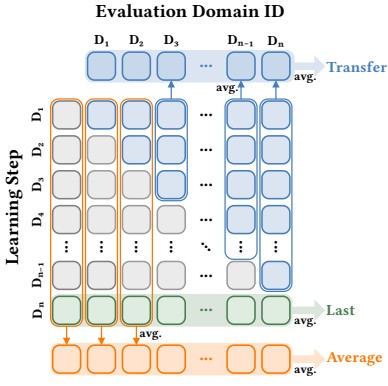

Figure 2: Metrics for X-TAIL setting.

green blocks, reflects the learner's adaptability to new domains. Additionally, the blue blocks in the upper-right matrix indicate the model's zero-shot performance on these domains before learning these domains. The average of these blocks, referred to as the *"Transfer"* metric, measures the extent to which the zero-shot ability is preserved throughout incremental learning.

# 4  Approach

## 4.1  Motivation

While CLIP demonstrates the generalizability of zero-shot classification, it still struggles in certain unfamiliar domains [21]. Leveraging CLIP's robust feature extraction capabilities, linear probe offers a straightforward approach to transfer the CLIP to these domains [13]. Among various linear solutions, Ridge Regression (RR) provides an effective classification strategy by mapping the image features onto one-hot-label targets [35]. Given a pre-trained CLIP image encoder $f_I$ and a dataset $D = \{(\mathbf{X}, \mathbf{Y})\}$, where $\mathbf{X}$ is the tensor of training images and $\mathbf{Y}$ is the matrix of corresponding one-hot labels, the predicted logits and the optimization problem are defined as:

$$\hat{\mathbf{y}} = \mathbf{X}_e \mathbf{W}, \quad \arg\min_{\mathbf{W}} \|\mathbf{Y} - \mathbf{X}_e \mathbf{W}\|_F^2 + \lambda \|\mathbf{W}\|_F^2 , \tag{1}$$

where $\mathbf{X}_e = f_I(\mathbf{X})$ denotes the CLIP extracted features, $\mathbf{W}$ is the classifier parameter, and $\lambda$ is the regularization parameter.

In the context of X-TAIL, the classifier needs to distinguish a wide range of classes from different domains. However, the extracted CLIP features of images from different domains suffer from certain cross-domain correlations, leading to limited domain discriminability. Based on Cover's theorem [36], one promising approach [37, 38, 39] to enhance the linear separability of features is to project the features into a higher-dimensional space via some non-linear projection function. We explore this non-linear projection function from two perspectives:

**Primal form ridge regression.** Following [38], we use a Randomly-initialized Hidden Layer (RHL) to project raw features to a higher dimensional space. By explicitly defining the projection function

as $\phi(\cdot)$, the classifier parameter is determined as follows:

$$\mathbf{W} = \left(\mathbf{\Phi}^\top \mathbf{\Phi} + \lambda \mathbf{I}\right)^{-1} \mathbf{\Phi}^\top \mathbf{Y}, \qquad (2)$$

where $\mathbf{\Phi} = \phi(\mathbf{X}_e)$. In this way, $\phi(\cdot)$ is fixed throughout the training process.

**Dual form ridge regression [40].** Instead of manually designing the projection function, we utilize the Kernel method [41] to implicitly define $\phi(\cdot)$ based on the inner-product nature of dual form ridge regression. Depending on the choice of kernel function, this approach allows for an infinite projection dimension, which is unachievable through any explicit definition. The dual form solution is defined as:

$$\boldsymbol{\alpha} = \left(\mathbf{K} + \lambda \mathbf{I}\right)^{-1} \mathbf{Y}, \qquad (3)$$

where $\mathbf{K} = \mathcal{K}(\mathbf{X}, \mathbf{X})$ denotes the covariance kernel matrix, and $\mathcal{K}(\cdot, \cdot)$ can be any positive-definite kernel function. The classification logits are derived as $\hat{\mathbf{y}} = \mathcal{K}(f_I(\mathbf{x}_{\text{test}}), \mathbf{X})\boldsymbol{\alpha}$. Throughout the paper, we use the Radial Basis Function (RBF) kernel [42] by default. The choice between primal and dual form ridge regression depends on whether the system is over-determined (more equations than unknowns) or under-determined (more unknowns than equations) [43]. Details on the relationships between primal and dual ridge regression can be found in Appendix B.

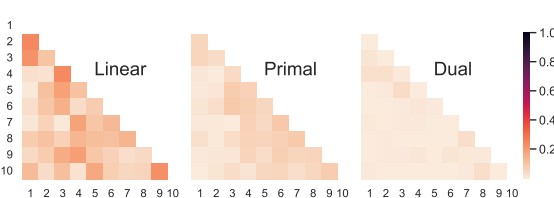

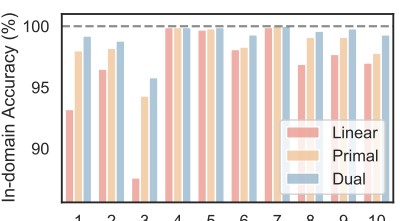

Figure 3: Pearson correlation coefficients (CCs) for 10 pairs of domain-prototypes.

Figure 4: Comparison of in-domain accuracy (%) on each domain with three classifiers.

To empirically verify whether these non-linear projections enhances the separability of CLIP features of images from different domains, we trained three types of classifiers (denoted as Linear, Primal and Dual, respectively) on 10 domains introduced in Sec. 3.2 jointly. To compare the standard linear regression form with aforementioned two approaches, we take the averaged weight vectors as the domain-prototypes and then calculate the inter-domain Pearson correlation coefficients (CCs) between 10 pairs of domain-prototypes. As shown in Fig. 3, the linear regression classifier exhibits high cross-domain correlations. By contrast, the RHL in the primal form significantly reduces these correlations. The implicit projection provided by the kernel trick in the dual form enables better disentangling of different domains. We further evaluate the in-domain accuracy, which represents the rate of correctly classifying images into the appropriate domains. Fig. 4 shows that the in-domain accuracy is negatively correlated to cross-domain correlations. Both primal and dual forms demonstrate certain improvements through the projection designs, allowing for accurate classification of images into their respective domains without domain identity hint.

## 4.2 Regression-based analytic incremental learning

Based on the projection approaches introduced above, we propose the Regression-based Analytic Incremental Learning (RAIL) method, which incorporates a ridge regression-based adapter and a training-free fusion module. The adapter progressively adapts the pre-trained CLIP to new domains, while the training-free fusion module preserves CLIP's zero-shot ability on unseen domains. An overview of RAIL is illustrated in Fig. 5. The pseudo-codes of both training and testing algorithms are provided in Appendix A.

### 4.2.1 RAIL-Adapter

In the context of CL, in which data arrives progressively, we extend both primal and dual ridge regression solutions to an incremental learning manner. Our solutions are identical to that obtained by joint training, which achieves absolute non-forgetting of learned knowledge. Let $D^{(n)} = \{\mathbf{X}^{(n)}, \mathbf{Y}^{(n)}\}$ represent the $n$-th training set and $D^{(1:n)} = \{\mathbf{X}^{(1:n)}, \mathbf{Y}^{(1:n)}\}$ represent the union of the training sets

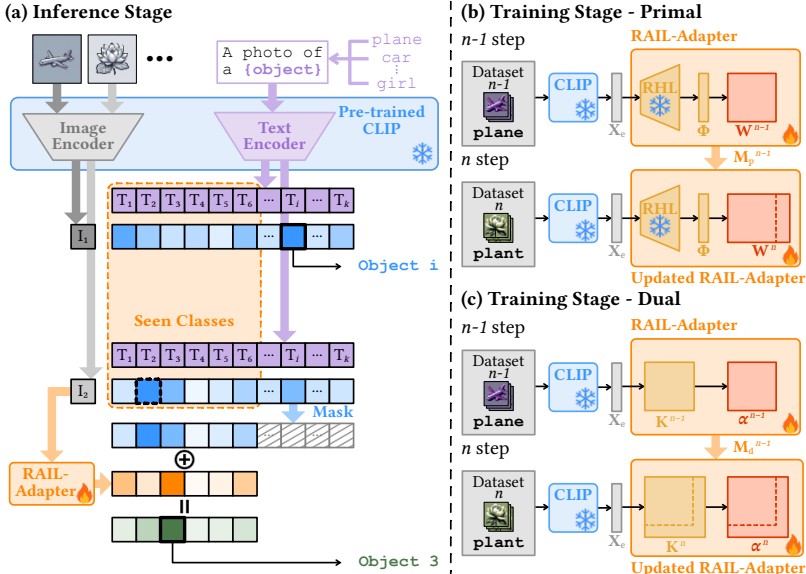

Figure 5: **RAIL Overview.** (a) During inference, the fusion module utilizes the **Zero-shot logits** to identify whether a test image is aligned with seen or unseen classes. If classified as a seen class, the **Fusion logits** combine the **RAIL-Adapter logits** and the **Zero-shot logits**; otherwise solely rely on the **Zero-shot logits**. (b) Primal: at the $n$-th learning step, features $\mathbf{X}_e$ extracted by CLIP's image encoder are projected to higher dimensional $\mathbf{\Phi}$ via RHL and then update the parameter $\mathbf{W}$ and memory $\mathbf{M}_p$ by Theorem 1. (c) Dual: features extracted by CLIP's image encoder update the kernel $\mathbf{K}$, parameter $\boldsymbol{\alpha}$, and memory $\mathbf{M}_d$ by Theorem 2.

from the first $n$ domains. At the $n$-th learning step, the optimization target for the joint training is expressed as

$$\underset{\mathbf{W}^{(n)}}{\arg\min} \left\| \mathbf{Y}^{(1:n)} - \mathbf{\Phi}^{(1:n)}\mathbf{W}^{(n)} \right\|_F^2 + \lambda \left\| \mathbf{W}^{(n)} \right\|_F^2, \tag{4}$$

where $\mathbf{\Phi}^{(1:n)} = \phi\left(f_I\left(\mathbf{X}^{(1:n)}\right)\right)$. The objective is to obtain $\mathbf{W}^{(n)}$ that satisfies Eqn. 4 without accessing data from the previous $n-1$ domains. For primal form, we propose to solve $\mathbf{W}^{(n)}$ recursively using $\mathbf{W}^{(n-1)}$ and a memory matrix $\mathbf{M}_p^{(n)}$. The solution is summarized as in Theorem 1.

**Theorem 1** *The parameter calculated by*

$$\mathbf{W}^{(n)} = \left[ \mathbf{W}^{(n-1)} - \mathbf{M}_p^{(n)}\mathbf{\Phi}^{(n)\top}\mathbf{\Phi}^{(n)}\mathbf{W}^{(n-1)} \quad \mathbf{M}_p^{(n)}\mathbf{\Phi}^{(n)\top}\mathbf{Y}^{(n)} \right] \tag{5}$$

*is an optimal solution to the optimization problem of joint training on all $n$ domains in Eqn. 4, where $\mathbf{M}_p^{(n)}$ is obtained by*

$$\mathbf{M}_p^{(n)} = \mathbf{M}_p^{(n-1)} - \mathbf{M}_p^{(n-1)}\mathbf{\Phi}^{(n)\top}\left(\mathbf{I} + \mathbf{\Phi}^{(n)}\mathbf{M}_p^{(n-1)}\mathbf{\Phi}^{(n)\top}\right)^{-1}\mathbf{\Phi}^{(n)}\mathbf{M}_p^{(n-1)}. \tag{6}$$

Similarly, the dual parameter $\boldsymbol{\alpha}^{(n)}$ satisfying Eqn. 4 can be obtained based on $\boldsymbol{\alpha}^{(n-1)}$, an updating kernel $\mathbf{K}^{(n)}$, and a memory matrix $\mathbf{M}_d^{(n)}$. We denote the matrix $\mathbf{C}^{(n)}$ as the concatenated one-hot label matrices of all $n$ domains. The solution is defined in Theorem 2.

**Theorem 2** *The parameter calculated by*

$$\boldsymbol{\alpha}^{(n)} = \left(\mathbf{K}^{(n)} + \lambda\mathbf{I}\right)^{-1}\mathbf{C}^{(n)} \tag{7}$$

*is an optimal solution to the optimization problem of joint training on all $n$ domains in Eqn. 4, where*

$$\mathbf{K}^{(n)} = \begin{bmatrix} \mathbf{K}^{(n-1)} & \mathcal{K}\left(\mathbf{X}_e^{(n)}, \mathbf{M}_d^{(n-1)}\right)^\top \\ \mathcal{K}\left(\mathbf{X}_e^{(n)}, \mathbf{M}_d^{(n-1)}\right) & \mathcal{K}\left(\mathbf{X}_e^{(n)}, \mathbf{X}_e^{(n)}\right) \end{bmatrix}, \quad \mathbf{C}^{(n)} = \begin{bmatrix} \mathbf{C}^{(n-1)} & \mathbf{0} \\ \mathbf{0} & \mathbf{Y}^{(n)} \end{bmatrix}, \tag{8}$$

*and the memory matrix is given by* $\mathbf{M}_d^{(n)} = \left[ \mathbf{M}_d^{(n-1)\top} \quad \mathbf{X}_e^{(n)\top} \right]^\top$.

At each incremental learning step, the kernel matrix $\mathbf{K}$ updates recursively along the main diagonal, preserving the correlations among class-prototypes from all domains. During testing, the kernel covariance between the feature of test image extracted by CLIP and memory matrix $\mathbf{M}_d$ is calculated to obtain the classification logits $\hat{\mathbf{y}} = \mathcal{K}\left(f_I\left(\mathbf{x}_{\text{test}}\right), \mathbf{M}_d\right)\boldsymbol{\alpha}$. Specifically, $\mathbf{M}_d$ dynamically updates according to the data stream via concatenated class-prototypes. These class-prototypes can be the feature embeddings $\mathbf{X}_e$ extracted by CLIP, K-means centroids, or Gaussian Mixture Model means. We use raw feature embeddings $\mathbf{X}_e$ by default, which is sufficient to validate our method. The complete proofs for both theorems are provided in Appendix C.

#### 4.2.2 RAIL-Fusion

Next, we introduce the fusion strategy, which leverages the refined knowledge on seen domains from the RAIL-Adapter while preserving CLIP's pre-trained knowledge on unseen domains. To distinguish data from different domains without any domain-identity hint, a common approach [18] is to compute domain centers from class-prototypes. The test image is first assigned to a specific domain based on the distances to these domain centers and then classified by a domain-specific classifier. However, this method fails when statistics for unseen domains are unavailable.

An alternative solution is to leverage CLIP's zero-shot ability to indicate the domain of the test image. Despite CLIP's strong generalization ability across domains, certain cross-domain errors (*i.e.,* misclassification into an incorrect domain) persist and do not diminish during incremental learning process. Therefore, the task of classifying images across seen domains is delegated to the RAIL-Adapter, which significantly reduces these errors, as discussed in Sec. 4.1. Consequently, CLIP's zero-shot ability is only leveraged to distinguish classes in unseen domains (*i.e.*, Out-Of-Distribution or OOD) from those in seen ones (*i.e.*, In-Distribution or ID) to maintain its performance on unseen domains. We summarize this approach as RAIL-Fusion, which combines both CLIP's zero-shot logits and RAIL-Adapter logits for prediction, regardless of whether the domain of the test image is seen or unseen.

Specifically, CLIP first makes a rough prediction based on its zero-shot logits, *i.e.,* the similarity scores between image embeddings and language embeddings from the cross-domain label set $C_N$:

$$\hat{\mathbf{y}}_{\text{zs}} = \text{Softmax}\left(f_I\left(\mathbf{x}_{\text{test}}\right) f_T\left(\text{Tokenizer}\left([P, C_N]\right)\right)^\top\right), \tag{9}$$

where $\hat{\mathbf{y}}_{\text{zs}}$ represents the zero-shot logits, $P$ denotes the pre-defined prompt template, and $f_T$ and $f_I$ are the CLIP text encoder and image encoder, respectively. The result determines whether the test image aligns with the seen classes (ID) that have been encountered during the incremental learning or with the unseen classes (OOD). If classified as ID, the RAIL-adapter refines the rough prediction using its incrementally learned knowledge. If classified as OOD, the rough prediction is taken as the final prediction, fully relying on CLIP's zero-shot ability. Notably, our fusion strategy guarantees that OOD images correctly classified by CLIP's zero-shot prediction will never be misclassified as ID, thereby *absolutely* preserving CLIP's zero-shot ability on unseen domains. In addition, to prevent the forgetting of the pre-trained knowledge of CLIP on domains with good zero-shot performance, we combine the zero-shot logits and RAIL-Adapter logits with a weighted sum:

$$\hat{\mathbf{y}}_{\text{fs}} = (1 - \beta)\,\hat{\mathbf{y}}_{\text{ad}} + \beta\hat{\mathbf{y}}_{\text{zs}}, \tag{10}$$

where $\hat{\mathbf{y}}_{\text{ad}}$ denotes the RAIL-Adapter logits and $\beta$ is the fusion ratio that adjusts the influence of zero-shot prediction on seen domains. The ablation study on $\beta$ is presented in Appendix E.2.

## 5  Experiments

We evaluate RAIL method under both X-TAIL and MTIL settings, as mentioned in Sec. 3.2. The learning order is set alphabetically: Aircraft, Caltech101, DTD, EuroSAT, Flowers, Food, MNIST, OxfordPet, StanfordCars, and SUN397. Additional experiments with a random order are provided in Tab. 3 in Appendix G. To ensure compatibility with different domains, we follow the common practice of sampling a 16-shot training set for each domain, while using the original test set for evaluation [21, 44, 45, 22, 46]. The implementation details are provided in Appendix D.

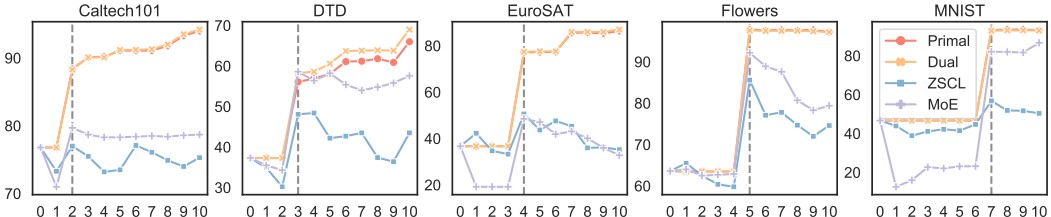

Figure 6: Accuracy (%) on five domains changes over all learning steps.

## 5.1 Comparison results

**Cross-domain task-agnostic incremental learning.** The performances averaged on 10 domains of RAIL and other baseline methods in the X-TAIL setting are presented in the *Average* column of Tab. 1. Specific performances on each domain are provided in the columns named after each respective domain. We evaluate RAIL in both primal and dual forms. *Zero-shot* indicates the zero-shot performance of the pre-trained CLIP model on each domain. *Fine-tune* denotes the performance of fine-tuning both CLIP image and text encoders with a joint dataset of all 10 domains, serving as a strong baseline for comparison.

Specifically, the primal-RAIL outperforms the previous best one with a $6.4\%$ improvement in *"Transfer"* accuracy, achieves an additional $7.7\%$ in *"Average"* accuracy, and gains an $8.6\%$ improvement in *"Last"* accuracy. The dual-RAIL further surpasses the primal one by $1.2\%$ in *"Average"* accuracy and $3.3\%$ in *"Last"* accuracy, while maintaining consistent *"Transfer"* accuracy due to the same fusion strategy. These results indicate that RAIL has more stable transfer performance and is more robust to catastrophic forgetting, effectively preserving both knowledge from new domains and pre-trained knowledge. We repeat the experiments with a random order and present the results in Tab. 3. RAIL consistently outperforms the baselines, reaffirming the previous conclusions.

We illustrate how accuracy changes on several example domains in Fig. 6. We observe that the accuracy of RAIL remains consistent with the zero-shot results before learning the corresponding domain. Furthermore, RAIL exhibits strong cross-domain discriminative capabilities. For example, once DTD is learned, learning further new domains does not affect the accuracy on DTD. Accuracy on certain domains, like Caltech101, even improves due to the fusion module's ability to reduce OOD errors by learning from more domains.

**Multi-domain task-incremental learning.** We follow the setting in [15] to evaluate our methods on the few-shot MTIL, comparing it against the performance of baselines reported in [15]. In this context, RAIL is reduced to the structure of multiple domain-specific classifiers trained on each domain separately. The domain-identity guides the test image to the corresponding classifier for within-domain prediction. The comparison results are shown in Tab. 2. RAIL consistently outperforms previous state-of-the-art approaches on all three metrics. These results demonstrate that the proposed non-linear projections significantly improve the separability of features extracted by CLIP. Consequently, the ridge regression-based classifier can effectively adapt the pre-trained model to new domains.

## 5.2 Discussion

**Regression targets.** For VLMs, aside from using one-hot labels as the regression targets $\mathbf{Y}$ in Eqn. 4, the text embeddings generated from class labels is also a viable option [39]. We compare the *"Last"* accuracy of 10 domains using the dual RAIL-Adapter with these two different regression targets as shown in Fig. 7a. The results indicate that training with one-hot labels surpasses its counterpart with text embeddings by an average of $3.8\%$. We emphasize that using text embeddings as targets is suboptimal compared to uniformly distributed one-hot labels. This effect is particularly notable in domains such as Aircraft, where the *"Last"* accuracy with one-hot label targets outperforms that with text embedding targets by $7.5\%$. We argue that insufficiently semantic class names (*e.g.*, "707-320") result in text embeddings that are not well-dispersed in the feature space, thus compromising the classification performance.

Table 1: Comparison of different CL methods on X-TAIL for each domain in terms of "Transfer", "Average", and "Last" scores (%). The best results are highlighted with **bold** style.

| Method | Aircraft | Caltech101 | DTD | EuroSAT | Flowers | Food101 | MNIST | Pets | Cars | Sun397 | Average |
|---|---|---|---|---|---|---|---|---|---|---|---|
| Zero-shot | 23.5 | 76.8 | 37.3 | 36.7 | 63.6 | 84.0 | 46.7 | 86.7 | 66.1 | 63.7 | 58.5 |
| Fine-tune | 39.6 | 93.3 | 68.2 | 89.2 | 95.4 | 85.5 | 95.1 | 84.4 | 77.4 | 72.4 | 80.1 |
| **Transfer** | | | | | | | | | | | |
| LwF [6] | – | 66.6 | 26.9 | 19.5 | 51.0 | 78.4 | 26.6 | 68.9 | 35.5 | 56.1 | 47.7 |
| WiSE-FT [47] | – | 70.1 | 31.9 | 25.3 | 56.3 | 79.8 | 29.9 | 74.9 | 45.6 | 56.8 | 52.3 |
| iCaRL [7] | – | 71.7 | 35.0 | 43.0 | 63.4 | 86.9 | 43.9 | 87.8 | 63.7 | 60.0 | 61.7 |
| ZSCL [10] | – | 73.3 | 32.6 | 36.8 | 62.1 | 83.8 | 42.1 | 83.6 | 56.5 | 60.2 | 59.0 |
| MoE-Adapter [15] | – | 71.0 | 34.9 | 19.2 | 63.0 | 86.6 | 20.0 | 87.2 | 63.7 | 58.6 | 56.0 |
| Primal-RAIL | – | **76.8** | **37.3** | 36.7 | **63.6** | 84.0 | **46.7** | 86.7 | **66.1** | **63.7** | **62.4** |
| Dual-RAIL | – | **76.8** | **37.3** | 36.7 | **63.6** | 84.0 | **46.7** | 86.7 | **66.1** | **63.7** | **62.4** |
| **Average** | | | | | | | | | | | |
| LwF | 24.7 | 79.7 | 38.3 | 36.9 | 63.9 | 81.0 | 36.5 | 71.9 | 42.7 | 56.7 | 53.2 |
| WiSE-FT | 27.1 | 76.5 | 40.9 | 31.3 | 68.7 | 81.6 | 31.4 | 74.7 | 51.7 | 58.4 | 54.2 |
| iCaRL | 25.4 | 72.1 | 37.5 | 51.6 | 65.1 | 87.1 | 59.1 | 88.0 | 63.7 | 60.1 | 61.0 |
| ZSCL | 36.0 | 75.0 | 40.7 | 40.5 | 71.0 | 85.3 | 46.3 | 83.3 | 60.7 | 61.5 | 60.0 |
| MoE-Adapter | 43.6 | 77.9 | 52.1 | 34.7 | 75.9 | 86.3 | 45.2 | 87.4 | 66.6 | 60.2 | 63.0 |
| Primal-RAIL | 42.4 | 89.8 | 55.7 | 68.5 | **84.0** | 83.3 | **65.3** | 85.8 | 67.9 | 64.5 | 70.7 |
| Dual-RAIL | **45.3** | **89.9** | **57.6** | **68.7** | 83.9 | 85.5 | 65.2 | **88.4** | **69.4** | **65.0** | **71.9** |
| **Last** | | | | | | | | | | | |
| LwF | 20.9 | 83.1 | 47.5 | 38.2 | 75.5 | 84.7 | 50.1 | 78.0 | 75.8 | 74.6 | 62.8 |
| WiSE-FT | 21.8 | 76.8 | 42.9 | 20.8 | 77.5 | 84.9 | 30.7 | 76.6 | 75.8 | 72.5 | 58.0 |
| iCaRL | 25.5 | 72.1 | 38.9 | 55.4 | 65.5 | 87.3 | 81.9 | 88.6 | 63.6 | 61.5 | 64.0 |
| ZSCL | 33.1 | 75.3 | 43.5 | 35.2 | 74.6 | **87.4** | 50.4 | 84.2 | 77.3 | 73.4 | 63.4 |
| MoE-Adapter | 43.2 | 78.7 | 57.6 | 32.8 | 79.4 | 86.0 | 86.7 | 87.8 | 78.2 | 74.2 | 70.5 |
| Primal-RAIL | 41.7 | 94.0 | 66.0 | 86.4 | **97.2** | 82.4 | **93.1** | 83.6 | 75.0 | 71.3 | 79.1 |
| Dual-RAIL | **45.3** | **94.2** | **69.0** | 87.0 | **97.2** | 87.2 | 93.0 | **92.4** | **82.5** | **76.3** | **82.4** |

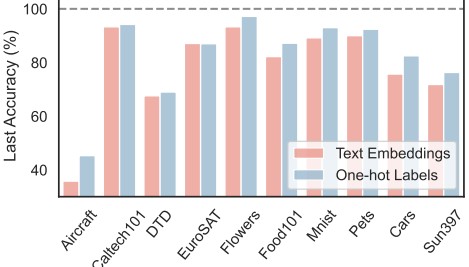

(a) Comparison of different regression targets.

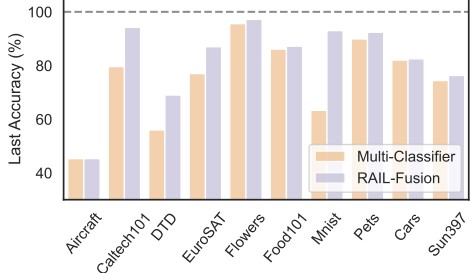

(b) Comparison of different fusion strategies.

Figure 7: Each bar indicates the *"Last"* accuracy (%) on each domain after the last learning step.

**Fusion strategies.** The CL setting associated with multiple domains is typically decomposed into two stages: domain-identity inference and in-domain prediction [19]. As discussed in Sec. 4.2.2, an intuitive strategy for distinguishing classes from different domains in X-TAIL is to utilize CLIP's zero-shot prediction as a domain indicator, which then cooperates with multiple domain-specific classifiers to perform classification within each distinct domain. We evaluate the effectiveness of the RAIL-Fusion against this strategy by comparing the *"Last"* accuracy across 10 domains using the dual RAIL-Adapter. The *"Transfer"* accuracy remains consistent between these two strategies.

As shown in Fig. 7b, the RAIL-Fusion strategy outperforms the multi-classifier approach in most domains by an average of 7.5%. This improvement is due to RAIL-Fusion's design, which focuses on distinguishing unseen classes (OOD domain) from seen classes (ID domain), rather than identifying specific domains (Sec. 4.2.2). This OOD detection design incrementally reduces errors as the number

Table 2: Comparison with state-of-the-art methods on 5-shot MTIL setting in terms of "Transfer", "Average", and "Last" scores (%). The best results are highlighted with **bold** style.

| Method | Aircraft | Caltech101 | CIFAR100 | DTD | EuroSAT | Flower | Food101 | MNIST | Pets | Cars | Sun397 | Average |
|---|---|---|---|---|---|---|---|---|---|---|---|---|
| Zero-shot | 24.3 | 88.4 | 68.2 | 44.6 | 54.9 | 71.0 | 88.5 | 59.6 | 89.0 | 64.7 | 65.2 | 65.3 |
| Fine-tune | 30.6 | 93.5 | 76.8 | 65.1 | 91.7 | 92.9 | 83.3 | 96.6 | 84.9 | 65.4 | 71.3 | 77.5 |
| **Transfer** | | | | | | | | | | | | |
| LwF | – | 72.1 | 49.2 | 35.9 | 44.5 | 41.1 | 66.6 | 50.5 | 69.0 | 19.0 | 51.7 | 50.0 |
| LwF-VR | – | 82.2 | 62.5 | 40.1 | 40.1 | 56.3 | 80.0 | 60.9 | 77.6 | 40.5 | 60.8 | 60.1 |
| WiSE-FT | – | 77.6 | 60.0 | 41.3 | 39.4 | 53.0 | 76.6 | 58.1 | 75.5 | 37.3 | 58.2 | 57.7 |
| ZSCL | – | 84.0 | 68.1 | **44.8** | 46.8 | 63.6 | 84.9 | 61.4 | 81.4 | 55.5 | 62.2 | 65.3 |
| MoE | – | 87.9 | **68.2** | 44.1 | 48.1 | 64.7 | **88.8** | **69.0** | **89.1** | 64.5 | 65.1 | 68.9 |
| Primal-RAIL | – | **88.4** | **68.2** | 44.6 | **54.9** | **71.0** | 88.5 | 59.6 | 89.0 | **64.7** | **65.2** | **69.4** |
| Dual-RAIL | – | **88.4** | **68.2** | 44.6 | **54.9** | **71.0** | 88.5 | 59.6 | 89.0 | **64.7** | **65.2** | **69.4** |
| **Average** | | | | | | | | | | | | |
| LwF | 23.5 | 77.4 | 43.5 | 41.7 | 43.5 | 52.2 | 54.6 | 63.4 | 68.0 | 21.3 | 52.6 | 49.2 |
| LwF-VR | 24.9 | 89.1 | 64.2 | 53.4 | 54.3 | 70.8 | 79.2 | 66.5 | 79.2 | 44.1 | 61.6 | 62.5 |
| WiSE-FT | 32.0 | 87.7 | 61.0 | 55.8 | 68.1 | 69.3 | 76.8 | 71.5 | 77.6 | 42.0 | 59.3 | 63.7 |
| ZSCL | 28.2 | 88.6 | 66.5 | 53.5 | 56.3 | 73.4 | 83.1 | 56.4 | 82.4 | 57.5 | 62.9 | 64.4 |
| MoE | 30.0 | 89.6 | **73.9** | 58.7 | 69.3 | 79.3 | 88.1 | **76.5** | 89.1 | 65.3 | **65.8** | 71.4 |
| Primal-RAIL | 32.9 | **94.5** | 69.9 | 58.1 | **71.8** | **84.4** | **88.5** | 70.4 | 89.0 | 66.1 | 65.7 | 71.9 |
| Dual-RAIL | **36.0** | 94.2 | 70.9 | **58.8** | 70.6 | 84.3 | **88.5** | 70.3 | **89.7** | **66.5** | **65.8** | **72.3** |
| **Last** | | | | | | | | | | | | |
| LwF | 22.1 | 58.2 | 17.9 | 32.1 | 28.1 | 66.7 | 46.0 | 84.3 | 64.1 | 31.5 | 60.1 | 46.5 |
| LwF-VR | 22.9 | 89.9 | 59.3 | 57.1 | 57.6 | 79.2 | 78.3 | 77.7 | 83.6 | 60.1 | 69.8 | 66.9 |
| WiSE-FT | 30.8 | 88.9 | 59.6 | 60.3 | 80.9 | 81.7 | 77.1 | **94.9** | 83.2 | 62.8 | 70.0 | 71.9 |
| ZSCL | 26.8 | 88.5 | 63.7 | 55.7 | 60.2 | 82.1 | 82.6 | 58.6 | 85.9 | 66.7 | 70.4 | 67.4 |
| MoE | 30.1 | 89.3 | **74.9** | 64.0 | **82.3** | 89.4 | 87.1 | 89.0 | 89.1 | 69.5 | **72.5** | 76.1 |
| Primal-RAIL | 32.9 | **95.1** | 70.3 | 63.2 | 81.5 | **95.6** | **88.5** | 89.7 | 89.0 | 72.5 | 71.0 | 77.2 |
| Dual-RAIL | **36.0** | 94.8 | 71.5 | **64.1** | 79.5 | 95.3 | **88.5** | 89.4 | **91.5** | **74.6** | 71.3 | **77.9** |

of ID domains grows, making it particularly effective in dynamically adapting to new domains. By contrast, using a domain indicator maintains a consistent level of cross-domain errors associated for each domain, leading to error propagation in the final prediction. This issue is especially critical for domains with low in-domain accuracy, where any misalignment between the domain indicator and the correct domain can significantly impact performance. These cross-domain errors are mitigated in the RAIL-Adapter thanks to its non-linear projection design.

# 6 Conclusion

In this work, we introduce the Cross-domain Task-Agnostic Incremental Learning (X-TAIL) to evaluate the preservation of pre-trained knowledge and cross-domain discriminative ability in a continual learning context. We introduce a novel CL approach, Regression-based Analytic Incremental Learning (RAIL), to improve the performance of pre-trained vision-language models on progressively incoming domains, while maintaining its zero-shot ability on unseen domains. We theoretically prove the absolute memorization on learned knowledge and show that the fusion module inherently avoids the forgetting of VLM's zero-shot ability. Comprehensive experiments on both existing and proposed settings empirically demonstrate the superiority of our method.

# 7 Acknowledgement

This research was supported by the National Natural Science Foundation of China (62306117) and the Guangzhou Basic and Applied Basic Research Foundation (2024A04J3681).

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

# Appendix

## A  Algorithm details

In this section, we summarize the training and testing procedures of RAIL in Algorithm 1 and 2, respectively.

---

**Algorithm 1** RAIL training

---

**Require:** $N$ domains $\{D^{(1)}, ..., D^{(N)}\}$, pre-trained CLIP model $\{f_I, f_T\}$
**Initialize:** Ridge regression parameter $\lambda$, random projection function $\phi(\cdot)$, kernel function $\mathcal{K}(\cdot, \cdot)$
  **Dataset 1 learning**
  **for** training batches in $D^{(1)}$ **do**
      Extract features $\mathbf{X}_e^{(1)} = f_I\left(\mathbf{X}^{(1)}\right)$
      **if Primal: then**
         Initialize the memory matrix $\mathbf{M}_p^{(1)}$ using Eqn. 15
         Obtain the primal parameter $\mathbf{W}^{(1)}$ using Eqn. 2
      **end if**
      **if Dual: then**
         Initialize the memory matrix $\mathbf{M}_d^{(1)}$, label matrix $\mathbf{C}^{(1)}$ & kernel $\mathbf{K}^{(1)}$ by Theorem 2
         Obtain the dual parameter $\boldsymbol{\alpha}^{(1)}$ using Eqn. 7
      **end if**
  **end for**
  **Incremental learning**
  **for** $D^{(n)}$ in $\{D^{(2)}, ..., D^{(N)}\}$ **do**
      **for** training batches in $D^{(1)}$ **do**
         Extract features $\mathbf{X}_e^{(n)} = f_I\left(\mathbf{X}^{(n)}\right)$
         **if Primal: then**
            Update the memory matrix $\mathbf{M}_p^{(n)}$ using Eqn.6
            Update the primal parameter $\mathbf{W}^{(n)}$ using Eqn. 5
         **end if**
         **if Dual: then**
            Update the memory matrix $\mathbf{M}_d^{(n)}$, label matrix $\mathbf{C}^{(n)}$ & kernel $\mathbf{K}^{(n)}$ by Theorem 2
            Update the dual parameter $\boldsymbol{\alpha}^{(n)}$ using Eqn. 7
         **end if**
      **end for**
  **end for**

---

**Algorithm 2** RAIL testing

---

**Require:** Test dataset $D_{\text{test}}$, CLIP model $\{f_I, f_T\}$, trained RAIL-Adapter $R_{ad}(\cdot)$, cross-domain label set $C_N$
  **for** $\mathbf{x}_{\text{test}} \in D_{\text{test}}$ **do**
      Obtain the rough prediction $y_{\text{zs}}^* = \arg\max \hat{\mathbf{y}}_{zs}$ by Eqn. 9
      **if** $y_{\text{zs}}^* \in C_L$ **then**
         Obtain RAIL-adapter logits $\hat{\mathbf{y}}_{ad} = R_{ad}\left(f_I\left(\mathbf{x}_{\text{test}}\right)\right)$
         Obtain predicted class based on fusion logits $y^* = \arg\max \hat{\mathbf{y}}_{fs}$ by Eqn. 10
      **else**
         Regard $y_{\text{zs}}^*$ as the final prediction
      **end if**
  **end for**

---

## B  Connection between primal & dual ridge regression

In this section, we introduce the connection between primal and dual forms of ridge regression.

Based on the identity $(\mathbf{P}^{-1}+\mathbf{B}^\top\mathbf{R}^{-1}\mathbf{B})^{-1}\mathbf{B}^\top\mathbf{R}^{-1}=\mathbf{P}\mathbf{B}^\top(\mathbf{B}\mathbf{P}\mathbf{B}^\top+\mathbf{R})^{-1}$, the solution of ridge regression is given by:

$$\mathbf{W} = (\mathbf{\Phi}^\top\mathbf{\Phi} + \lambda\mathbf{I}_d)^{-1}\mathbf{\Phi}^\top\mathbf{Y} = \mathbf{\Phi}^\top(\mathbf{\Phi}\mathbf{\Phi}^\top + \lambda\mathbf{I}_n)^{-1}\mathbf{Y}, \tag{11}$$

where the former solution is based on the outer-product of data and the latter one is based on the inner-product of data. Parameter $\mathbf{W}$ can thus be rewritten as: $\mathbf{W} = \mathbf{\Phi}^\top\boldsymbol{\alpha}$ with $\boldsymbol{\alpha} = (\mathbf{\Phi}\mathbf{\Phi}^\top + \lambda\mathbf{I}_n)^{-1}\mathbf{Y}$. In this way, the solution $\mathbf{W}$ is interpreted to lie in the span of the sample-cases, even if the dimensionality of the projected features $\mathbf{\Phi}$ is larger than the number of samples.

Utilizing the kernel method, we never actually require access to the explicit features $\mathbf{\Phi}$, which could be of indefinite dimensions. We obtain the prediction with given data $\mathbf{x}$ by projecting it onto the solution $\mathbf{W}$,

$$\hat{\mathbf{y}} = \phi\left(\mathbf{x}\right)\mathbf{W} = \phi\left(\mathbf{x}\right)\mathbf{\Phi}^\top(\mathbf{\Phi}\mathbf{\Phi}^\top + \lambda\mathbf{I}_n)^{-1}\mathbf{Y} = \mathcal{K}\left(\mathbf{x}, \mathbf{X}\right)\boldsymbol{\alpha}, \tag{12}$$

where $\mathcal{K}\left(\mathbf{x}_i, \mathbf{x}_j\right) = \phi\left(\mathbf{x}_i\right)^\top\phi\left(\mathbf{x}_j\right)$. What we require here is the choice of kernel function $\mathcal{K}\left(\cdot, \cdot\right)$ instead of explicitly defining the projection function.

## C  Proof of Theorems

In this section, we provide two mathematical proofs for both Theorem 1 and Theorem 2.

First, we prove the Theorem 1 from the solution of joint training on $n$ datasets with primal ridge regression:

$$\mathbf{W}^{(n)} = \left(\mathbf{\Phi}^{(1:n)\top}\mathbf{\Phi}^{(1:n)} + \lambda\mathbf{I}\right)^{-1}\mathbf{\Phi}^{(1:n)\top}\mathbf{Y}^{(1:n)}. \tag{13}$$

By decoupling the $n$-th data from previous datasets, the $\mathbf{W}^{(n)}$ can be written as:

$$\mathbf{W}^{(n)} = \left(\begin{bmatrix}\mathbf{\Phi}^{(1:n-1)\top} & \mathbf{\Phi}^{(n)\top}\end{bmatrix}\begin{bmatrix}\left(\mathbf{\Phi}^{(1:n-1)}\right) \\ \left(\mathbf{\Phi}^{(n)}\right)\end{bmatrix} + \lambda\mathbf{I}\right)^{-1}\begin{bmatrix}\mathbf{\Phi}^{(1:n-1)\top} & \mathbf{\Phi}^{(n)\top}\end{bmatrix}\begin{bmatrix}\mathbf{Y}^{(1:n-1)} & \mathbf{0} \\ \mathbf{0} & \mathbf{Y}^{(n)}\end{bmatrix}$$
$$= \left(\mathbf{\Phi}^{(1:n-1)\top}\mathbf{\Phi}^{(1:n-1)} + \lambda\mathbf{I} + \mathbf{\Phi}^{(n)\top}\mathbf{\Phi}^{(n)}\right)^{-1}\begin{bmatrix}\mathbf{\Phi}^{(1:n-1)\top}\mathbf{Y}^{(1:n-1)} & \mathbf{\Phi}^{(n)\top}\mathbf{Y}^{(n)}\end{bmatrix}. \tag{14}$$

We introduce the memory matrix as in the following definition:

$$\mathbf{M}_p^{(n)} = \left(\mathbf{\Phi}^{(1:n)\top}\mathbf{\Phi}^{(1:n)} + \lambda\mathbf{I}\right)^{-1}, \tag{15}$$

which is the matrix inversion term of Eqn 13.

Noticing that $\mathbf{M}_p^{(n-1)} = \left(\mathbf{\Phi}^{(1:n-1)\top}\mathbf{\Phi}^{(1:n-1)} + \lambda\mathbf{I}\right)^{-1}$, by Woodbury matrix identity where $(\mathbf{A} + \mathbf{U}\mathbf{C}\mathbf{V})^{-1} = \mathbf{A}^{-1} - \mathbf{A}^{-1}\mathbf{U}\left(\mathbf{C}^{-1} + \mathbf{V}\mathbf{A}^{-1}\mathbf{U}\right)^{-1}\mathbf{V}\mathbf{A}^{-1}$ and treating $\mathbf{M}_p^{(n-1)}$ as $\mathbf{A}^{-1}$, the memory at $n$-th step can be further defined as a recursive solution:

$$\mathbf{M}_p^{(n)} = \mathbf{M}_p^{(n-1)} - \mathbf{M}_p^{(n-1)}\mathbf{\Phi}^{(n)\top}\left(\mathbf{I} + \mathbf{\Phi}^{(n)}\mathbf{M}_p^{(n-1)}\mathbf{\Phi}^{(n)\top}\right)^{-1}\mathbf{\Phi}^{(n)}\mathbf{M}_p^{(n-1)}. \tag{16}$$

Thus, the parameter $\mathbf{W}^{(n)}$ is derived as

$$\mathbf{W}^{(n)} = \begin{bmatrix}\mathbf{M}_p^{(n)}\mathbf{\Phi}^{(1:n-1)\top}\mathbf{Y}^{(1:n-1)} & \mathbf{M}_p^{(n)}\mathbf{\Phi}^{(n)\top}\mathbf{Y}^{(n)}\end{bmatrix}. \tag{17}$$

Denotes the left submatrix $\mathbf{M}_p^{(n)}\mathbf{\Phi}^{(1:n-1)\top}\mathbf{Y}^{(1:n-1)}$ as $\mathbf{H}$. By substituting Eqn 16 into 17,

$$\mathbf{H} = \mathbf{W}^{(n-1)} - \mathbf{M}_p^{(n-1)}\mathbf{\Phi}^{(n)\top}\left(\mathbf{I} + \mathbf{\Phi}^{(n)}\mathbf{M}_p^{(n-1)}\mathbf{\Phi}^{(n)\top}\right)^{-1}\mathbf{\Phi}^{(n)}\mathbf{W}^{(n-1)}. \tag{18}$$

Based on the identity of $(\mathbf{I} + \mathbf{P})^{-1} = \mathbf{I} - (\mathbf{I} + \mathbf{P})^{-1}\mathbf{P}$, it is further derived as:

$$\mathbf{H} = \mathbf{W}^{(n-1)} - \mathbf{M}_p^{(n)}\boldsymbol{\Phi}^{(n)\top}\boldsymbol{\Phi}^{(n)}\mathbf{W}^{(n-1)}. \tag{19}$$

Thus,

$$\mathbf{W}^{(n)} = \left[\mathbf{W}^{(n-1)} - \mathbf{M}_p^{(n)}\boldsymbol{\Phi}^{(n)\top}\boldsymbol{\Phi}^{(n)}\mathbf{W}^{(n-1)} \quad \mathbf{M}_p^{(n)}\boldsymbol{\Phi}^{(n)\top}\mathbf{Y}^{(n)}\right]. \tag{20}$$

The Theorem 1 is proved.

Next, we prove the Theorem 2 as follows. We use the few-shot features $\mathbf{X}_e$ as the class-prototypes as default. The solution of joint training on $n$ datasets with dual form ridge regression is shown as:

$$\boldsymbol{\alpha}^{(n)} = \left(\mathcal{K}\left(\mathbf{X}_e^{(1:n)}, \mathbf{X}_e^{(1:n)}\right) + \lambda\mathbf{I}\right)^{-1}\mathbf{Y}^{(1:n)}. \tag{21}$$

Define the memory matrix $\mathbf{M}_d$ as the concatenation of class-prototypes from learned domains, the memory matrix at $n$-th step is obtained by:

$$\mathbf{M}_d^{(n)} = \left[\mathbf{M}_d^{(n-1)} \quad \mathbf{X}_e^{(n)}\right]. \tag{22}$$

The kernel matrix at $n$-th step for $\boldsymbol{\alpha}^{(n)}$ can be partitioned as:

$$\begin{aligned}
\mathbf{K}^{(n)} &= \mathcal{K}\left(\mathbf{X}_e^{(1:n)}, \mathbf{X}_e^{(1:n)}\right) \\
&= \begin{bmatrix} \mathcal{K}\left(\mathbf{X}_e^{(1:n-1)}, \mathbf{X}_e^{(1:n-1)}\right) & \mathcal{K}\left(\mathbf{X}_e^{(n)}, \mathbf{M}_d^{(n-1)}\right)^{\top} \\ \mathcal{K}\left(\mathbf{X}_e^{(n)}, \mathbf{M}_d^{(n-1)}\right) & \mathcal{K}\left(\mathbf{X}_e^{(n)}, \mathbf{X}_e^{(n)}\right) \end{bmatrix} \\
&= \begin{bmatrix} \mathbf{K}^{(n-1)} & \mathcal{K}\left(\mathbf{X}_e^{(n)}, \mathbf{M}_d^{(n-1)}\right)^{\top} \\ \mathcal{K}\left(\mathbf{X}_e^{(n)}, \mathbf{M}_d^{(n-1)}\right) & \mathcal{K}\left(\mathbf{X}_e^{(n)}, \mathbf{X}_e^{(n)}\right) \end{bmatrix}.
\end{aligned} \tag{23}$$

It indicates that the kernel matrix updates recursively along the diagonal by kernel matrices of the intra-domain covariance within $\mathbf{X}_e^{(n)}$ and the inter-domain covariance between $\mathbf{X}_e^{(n)}$ and the memory $\mathbf{M}_d^{(n-1)}$. The kernel matrix $\mathbf{K}$ therefore memorizes all the covariance information of learned domains. We further denote $\mathbf{Y}^{(1:n)}$ as $\mathbf{C}^{(n)}$, which is updated by:

$$\mathbf{C}^{(n)} = \begin{bmatrix} \mathbf{C}^{(n-1)} & \mathbf{0} \\ \mathbf{0} & \mathbf{Y}^{(n)} \end{bmatrix}, \tag{24}$$

where the matrix $\mathbf{Y}^{(n)}$ consists of one-hot labels that are disjoint with those in previous $n-1$ domains. In this way, the parameter $\boldsymbol{\alpha}^{(n)}$ is solved by updating $\mathbf{K}^{(n)}$ and $\mathbf{C}^{(n)}$, resulting in an identical solution to the one of joint learning in Eqn. 21. Thus, the Theorem 2 is proved.

## D   Implementation details

In this section, we introduce the implementation details, including model configuration, hardware setup, and hyperparamter selection. We use the pre-trained CLIP [13] model of the ViT-B/16 image encoder [48]. All the results are conducted on Ubuntu 20.04 with Intel Core i9-13900K CPU with a single RTX 4090Ti GPU by the average of 3 runs. We conduct a grid search for the regularization parameter $\lambda$ over the range $10^{-6}, 10^{-5}, ..., 1$ and the RBF kernel bandwidth over the range $10^{-6}, 10^{-5}, ..., 10$. The optimal values are determined by minimizing the regression error on the validation set of the first domain, without access to future domains. These parameters are then fixed for all subsequent learning steps. We use the simplest prompt template *"A photo of a {}."* for generalization across different domains.

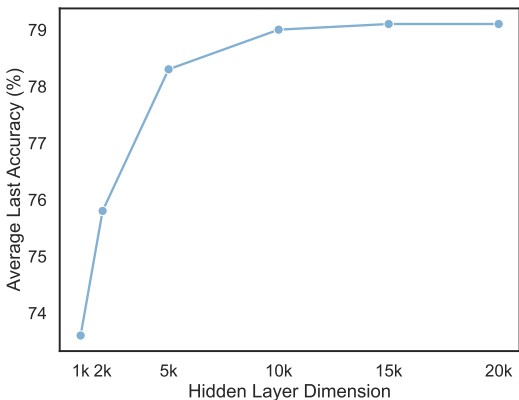

Figure 8: RHL dimension vs. *"Last"* accuracy (%) averaged on 10 domains.

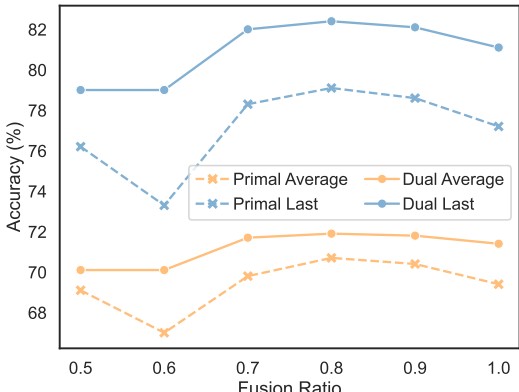

Figure 9: Fusion ratio vs. *"Average"* and *"Last"* accuracy (%) averaged on 10 domains.

# E    Ablation studies

In this section, we conduct two ablation studies to observe the average performance on 10 domains *w.r.t.* the hidden dimension of RHL and the fusion ratio, respectively.

## E.1    RHL dimension

We first ablate the hidden dimension of RHL in primal RAIL over the *"Last"* accuracy in Fig. 8. It is evident that an increase in the RHL dimension correlates with improved adapter's accuracy. Specifically, increasing the dimension from 1k to 10k leads to notable improvements (from 73.6% to 79.0%). However, beyond the 10k threshold, the gain in accuracy becomes saturated. The dimensions of 15k and 20k result the same performance of 79.1%. Considering the computational cost associated with higher dimensions, we set the RHL dimension to 15k as default in the experiments.

## E.2    Fusion ratio

Additionally, we conduct an ablation study of fusion ratio $\beta$ in terms of *"Average"* and *"Last"* accuracy. The *"Transfer"* accuracy is not considered here since $\beta$ does not influence it. From Fig. 9, we observe that the best ratio is $0.8$ for both *"Average"* and *"Last"* scores. Note that when $\beta$ equals to 1, the performance on seen domains fully relies on the RAIL-Adapter. The *"Last"* accuracy with $\beta = 0.8$ surpasses the one of pure RAIL-Adapter performance ($\beta = 1$) by 1.9% and 1.3% in the primal and dual RAIL, respectively. This result verifies our claim that the cooperation with CLIP's generalization ability can preserve the performance on its confident domains (already having good zero-shot performance) and avoid overfitting on limited domain exemplars.

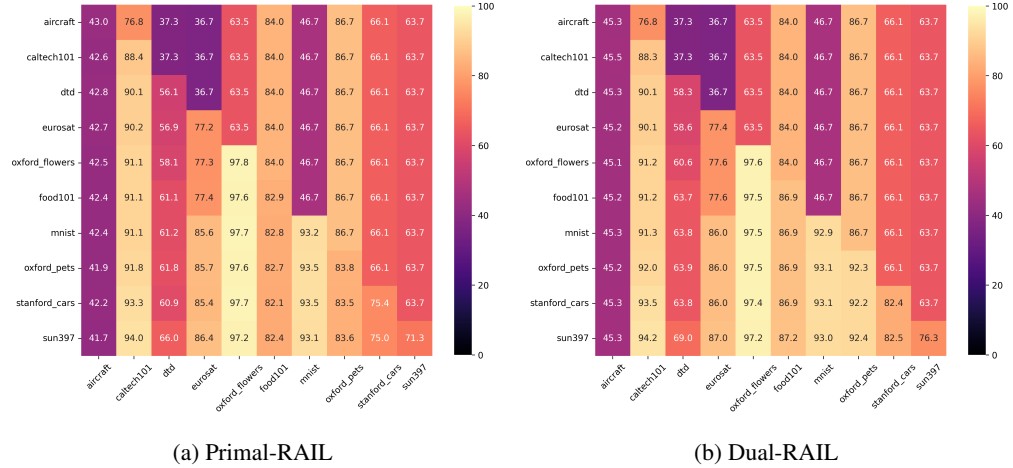

|                | (a) Primal-RAIL | (b) Dual-RAIL |
| :------------: | :-------------: | :-----------: |

Figure 10: Accuracy (%) of Primal-RAIL and Dual-RAIL in the X-TAIL setting with order-I. Each row denotes the performance on every domain after learning on one domain.

## F    Detailed performance of RAIL with order-I

In this section, we visualize the performance on each domain after every learning step in Fig. 10. The upper-diagonal represents the performance before learning on the corresponding domain, which remains consistent with the zero-shot performance thanks to RAIL's absolute memorization of the zero-shot ability on unseen domains. On the other hand, the performance after learning a specific domain (lower-diagonal) does not degrades. Performance on some domains even improves with the learning step (e.g., Caltech-101), benefiting from our RAIL fusion module design, which reduces OOD errors as more domains are learned, thereby enhancing overall accuracy.

## G    Comparison of different methods on X-TAIL with order II.

In this section, we compare different methods in X-TAIL setting with a random order: StanfordCars, Aircraft, OxfordPet, Food, SUN397, MNIST, Flowers, DTD, Caltech101, EuroSAT. As shown in Tab. 3, our method again outperforms previous methods on all metrics, reinforcing the conclusions presented in Sec. 5.1.

## H    Limitation

RAIL exhibits its superior performance and efficiency for transferring pre-trained VLMs to various domains. A limitation in this approach is that the pre-trained VLM remains fixed, preventing any improvement in its feature extraction capability throughout the incremental learning process. A promising direction for future work is to adjust the pre-trained encoder according to new data with low computational cost, thus further boosting RAIL's performance while maintain its efficiency. Beside, extending RAIL to encompass additional downstream tasks of VLMs, such as image segmentation, can broaden its applicability and enhance its utility in more complex visual understanding scenarios.

Table 3: Comparison of different CL methods on X-TAIL for each domain with order II in terms of "Transfer", "Average", and "Last" scores (%). The best results are highlighted with **bold** style.

| Method | Cars | Aircraft | Pets | Food | SUN397 | MNIST | Flower | DTD | Caltech101 | EuroSAT | Average |
|---|---|---|---|---|---|---|---|---|---|---|---|
| Zero-shot | 66.1 | 23.5 | 86.7 | 84.0 | 63.7 | 46.7 | 63.6 | 37.3 | 76.8 | 36.7 | 58.5 |
| Fine-tune | 77.4 | 39.6 | 84.4 | 85.5 | 72.4 | 95.1 | 95.4 | 68.2 | 93.3 | 89.2 | 80.1 |
| **Transfer** | | | | | | | | | | | |
| LwF | – | 20.0 | 74.1 | 79.6 | 58.1 | 34.1 | 48.9 | 27.7 | 64.4 | 15.1 | 46.9 |
| WiSE-FT | – | 21.3 | 79.5 | 83.3 | 61.0 | 39.9 | 56.5 | 29.6 | 68.0 | 20.8 | 51.1 |
| ZSCL | – | 23.0 | 84.3 | 87.2 | 63.0 | 42.1 | 65.2 | 34.6 | 71.4 | **40.9** | 56.9 |
| MoE-Adapter | – | 17.1 | **87.2** | **87.5** | 58.4 | 12.6 | **65.5** | 35.9 | 70.0 | 17.9 | 50.2 |
| Primal-RAIL | – | **23.5** | 86.7 | 84.0 | **63.7** | **46.7** | 63.5 | **37.3** | **76.8** | 36.7 | **57.7** |
| Dual-RAIL | – | **23.5** | 86.7 | 84.0 | **63.7** | **46.7** | 63.5 | **37.3** | **76.8** | 36.7 | **57.7** |
| **Average** | | | | | | | | | | | |
| LwF | 49.0 | 27.4 | 69.7 | 83.0 | 65.7 | 42.2 | 63.5 | 33.1 | 68.5 | 17.5 | 52.0 |
| WiSE-FT | 57.9 | 29.6 | 77.8 | 85.4 | 68.0 | 51.6 | 69.3 | 35.5 | 71.0 | 23.0 | 56.9 |
| ZSCL | 74.4 | 36.4 | 86.7 | **88.7** | 68.9 | 50.0 | 75.1 | 40.1 | 72.5 | **43.7** | 63.6 |
| MoE-Adapter | 74.4 | 38.6 | 87.7 | 87.3 | 67.9 | 50.6 | 76.5 | 43.7 | 72.3 | 18.8 | 61.8 |
| Primal-RAIL | 77.9 | 40.4 | 85.6 | 83.3 | 68.3 | 62.2 | 76.6 | 45.8 | **80.4** | 41.7 | 66.2 |
| Dual-RAIL | **82.3** | **43.4** | **90.4** | 86.0 | **71.0** | **62.8** | **76.7** | **46.7** | 80.3 | 41.7 | **68.1** |
| **Last** | | | | | | | | | | | |
| LwF | 29.6 | 17.5 | 63.0 | 83.8 | 67.7 | 44.9 | 79.3 | 44.8 | 84.6 | 39.0 | 55.4 |
| WiSE-FT | 46.1 | 23.5 | 71.3 | 85.7 | 70.2 | 59.1 | 85.5 | 47.9 | 82.4 | 42.8 | 61.5 |
| ZSCL | 71.7 | 35.3 | 86.5 | **89.2** | 71.8 | 52.3 | 89.8 | 52.0 | 77.1 | 68.4 | 69.4 |
| MoE-Adapter | 75.1 | 41.1 | 87.9 | 87.1 | 74.1 | 89.7 | 92.6 | 61.2 | 81.0 | 27.4 | 71.7 |
| Primal-RAIL | 77.7 | 41.9 | 86.1 | 83.3 | 71.8 | 91.6 | 97.3 | 66.4 | **94.8** | 86.9 | 79.8 |
| Dual-RAIL | **82.3** | **45.8** | **92.0** | 87.1 | **76.3** | **93.1** | **97.4** | **69.1** | 94.4 | **87.1** | **82.5** |

