# OpenReview forum: "Advancing Cross-domain Discriminability in Continual Learning of Vision-Language Models"
_NeurIPS.cc/2024/Conference — NeurIPS 2024 poster_

### Official Review · Reviewer_ZU8x · 2024-07-06

**Soundness:** 2
**Presentation:** 2
**Contribution:** 2
**Rating:** 5
**Confidence:** 4

**Summary:**

The paper points out that current VLM-based incremental learning tasks face the issue of text being limited to the corresponding task. It aims to propose a method that can achieve better incremental classification performance on a broader range of texts.

Specifically, the paper proposes Regression-based Analytic IncrementalLearning (RAIL), which utilizes a recursive ridge regression-based adapter to learn from a sequence of domains in a non-forgetting manner.

**Strengths:**

1. This work introduces a Cross-domain Task-Agnostic Incremental Learning (X-TAIL) setting, which evaluates the performance on a broader range of texts about VLM's continual learning ability.
2. The paper proposes a framework RAIL, which can incrementally transfer a pre-trained VLM to multiple domains while evaluating the model's performance on both seen and unseen domains.

**Weaknesses:**

1. Compared to existing methods, one dataset is missing. Why was CIFAR-100 removed from the forgetting benchmark? Is it because the method performs poorly on this dataset?
2. Determining whether a sample is OOD relies on learned class labels, which inevitably use label (text) information. I am concerned that this could lead to label leakage.
3. Figure 2d's graph is consistent with ZSCL, only styled differently, which I find unnecessary. It may not highlight the novelty of the X-TAIL setting.

**Questions:**

1. Additionally, during the fine-tuning stage, is each task fine-tuned using only the text corresponding to that task to ensure no label leakage occurs during the training of each task?
2. Why is the setting for Order2 different from existing methods? I am concerned about the actual performance of this method.
3. How does Formula 1 compute one-hot with embedding, given that $Y$ and $X_e$ are of different dimensions? Could you clarify the content of the formula clearly?

**Limitations:**

I am primarily concerned about the application scenarios and practical significance of the proposed setting, as well as the insufficient experimental validation.

---

> ### Author Rebuttal · Authors · 2024-08-04
>
> Thank you for your detailed feedback, which we believe will improve our final manuscript. Please see our responses to the helpful points raised in your review below:
> ***
>
> >***W1** Compared to existing methods, one dataset is missing. Why was CIFAR-100 removed from the forgetting benchmark? Is it because the method performs poorly on this dataset?*
>
> **A1** We appreciate your concern regarding the exclusion of CIFAR-100 from the X-TAIL setting.
> We have added experiments of including CIFAR-100 dataset in the X-TAIL setting. Due to the space limitation, please refer to the **Experiment 3 in the general response**. Our methods surpass existing SOTA methods on CIFAR-100 across all metrics.
>
> However, the primary purpose of X-TAIL is to evaluate a model's cross-domain discriminative capabilities while transferring continually to distinct domains. **CIFAR-100 was excluded because it includes many classes that overlap with those in other domains**; for instance, the 'forest' class in CIFAR-100 overlaps with 'broadleaf forest' in EuroSAT. During testing, if an image from EuroSAT labeled as 'broadleaf forest' is classified simply as 'forest,' this would be factually correct but considered incorrect in the evaluation metrics. To prevent the redundancy of learning overlapping classes and to maintain the integrity of the setting, CIFAR-100 was not included in X-TAIL. We will include a detailed explanation in our manuscript for clarification.
> ***
> >***W2** Determining whether a sample is OOD relies on learned class labels, which inevitably use label (text) information. I am concerned that this could lead to label leakage.*
>
> **A2** We would like to respectfully clarify that our method does **NOT** lead to label leakage. We determine whether a sample is OOD (*i.e.*, belonging to unlearned domain) during the testing process by checking if the CLIP zero-shot predicted class belongs to the set of classes learned by the adapter. If the zero-shot result of CLIP falls within the recorded set, the test image is classified as ID; otherwise, it is classified as OOD. This process does not involve any exposure to the labels or domains of the test images, **thus does NOT lead to label leakage.**
>
> We will emphasize this aspect more explicitly in the final version to ensure there is no confusion.
> ***
> >***W3** Figure 2d's graph is consistent with ZSCL, only styled differently, which I find unnecessary. It may not highlight the novelty of the X-TAIL setting.*
>
> **A3** We acknowledge that the metric presented in this figure is not a novelty of our work but rather an essential aspect of understanding the metrics involved in the X-TAIL setting. The styling similarity to ZSCL is intended solely for clarity and consistency in presentation, not as a highlight of novelty. Our intent is to facilitate reader understanding of the setting's metrics, as evidenced by the confusion expressed by Reviewer ThWm. We will add another citation to ZSCL in the caption of Figure 2 in the final version for further clarification.
> ***
> >***Q1** Additionally, during the fine-tuning stage, is each task fine-tuned using only the text corresponding to that task to ensure no label leakage occurs during the training of each task?*
>
> **A4** During the fine-tuning stage, we ensure that labels (texts) from future tasks are not accessible to current task, thereby preventing any label leakage problem.
> ***
> >***Q2** Why is the setting for Order2 different from existing methods? I am concerned about the actual performance of this method.*
>
> **A5** The order 2 of X-TAIL is randomly shuffled. In response to your concerns, we have conducted additional experiments as outlined in **Experiment 4 in the general response**, following the Order 2 sequence in ZSCL paper[1]. Our methods consistently surpass existing methods, similar to the results observed in the two orders discussed in the manuscript. This demonstrates the robustness of our methods to variations in task order.
> ***
> >***Q3** How does Formula 1 compute one-hot with embedding, given that Y and Xe are of different dimensions? Could you clarify the content of the formula clearly?*
>
> **A6** We apologize for the confusion regarding Formula 1. This was indeed a typo in our manuscript. $\mathbf{X}_e$ should be multiplied by a weight matrix $\mathbf{W}$ to map it to the same dimension as $\mathbf{Y}$. We will correct this formula according to your observation and the suggestion provided by Reviewer Tz45.
> ***
> >***Limitation** I am primarily concerned about the application scenarios and practical significance of the proposed setting, as well as the insufficient experimental validation.*
>
> **A7** The X-TAIL setting primarily addresses a significant limitation in the existing continual learning scenarios for Vision-Language Models, specifically within the MTIL setting where specifying the domain information of test images is necessary. The contribution of our setting is also acknowledged by Reviewer Tz45. To further validate the effectiveness of our method, we have conducted **additional experiments in the general response**. If you have any questions about our setting, we would be glad to provide additional clarification during the discussion.
> ***
> Based on these additional results and clarifications, we hope you could consider increasing your score in support of this work. If not, could you kindly let us know what additionally needs to be done in your assessment to make this work ready for publication?
> ***
> ***Reference***
>
> [1] Zheng, Zangwei, et al. "Preventing zero-shot transfer degradation in continual learning of vision-language models." *ICCV*. 2023.

---

> > ### Comment · Reviewer_ZU8x · 2024-08-10
> >
> > Thank you for the response. It has addressed most of my concerns and I will raise my rating to 5.

---

> > > ### Author Response · Authors · 2024-08-10
> > > **Thank you for the response**
> > >
> > > Thank you for reading our response and increasing your score to support us! We are glad to hear that the response addressed your concerns.

---

### Official Review · Reviewer_rqEC · 2024-07-10

**Soundness:** 3
**Presentation:** 4
**Contribution:** 4
**Rating:** 7
**Confidence:** 4

**Summary:**

This paper proposes a Regression-based Analytic Incremental Learning (RAIL). It utilizes a recursive ridge regression-based adapter to learn from a sequence of domains in a non-forgetting manner and decouple the cross-domain correlations by projecting features to a higher-dimensional space. Additionally, the paper introduces Cross-domain Task-Agnostic Incremental Learning (X-TAIL) setting. The paper theoretically proves RAIL’s absolute memorization on incrementally learned domains. Experiment results affirm RAIL’s state-of-the-art performance in both X-TAIL and existing Multi-domain Task-Incremental Learning settings.

**Strengths:**

1. The proposed RAIL adopts traditional machine learning techniques (e.g., Primal and Dual forms) for dealing with continual learning is novel.
2. The idea of RAIL’s absolute memorization based on analytic techniques is very appealing. Upon a good pre-trained network, the forgetting problem no longer exists, and this is very rare in the continual learning community.
3. The paper is easy to follow overall.
4. It was difficult to handle CIL problems in multi-modal CIL. This paper extends the continual learning scenario from task incremental to class incremental without task ID. This is a good contribution.
5. Experiments and settings are overall well formulated.

**Weaknesses:**

1. Could you explain the main difference between primal and dual forms in the CIL problem? It could be quite diffcult in this community to understand.
2. Experiemnts seems a little bit thin in the manucript. Perhaps try to move some of the experiments in the appendix to the main content.

**Questions:**

Please refer to the Weakness.

---

> ### Author Rebuttal · Authors · 2024-08-04
>
> We appreciate the reviewer's positive feedback on our work, and further thank the reviewer for finding our work novel in adopting traditional machine learning techniques for continual learning and appealing in its approach to absolute memorization based on analytic techniques. Below, we address the reviewer's concerns in turn:
> ***
> >***W1** Could you explain the main difference between primal and dual forms in the CIL problem? It could be quite difficult in this community to understand.*
>
> **A1** In the proposed method, the primary distinction between the primal and dual forms lies in the differences between primal ridge regression and dual ridge regression: the former **explicitly** projects CLIP extracted features into a higher dimensional space, while the latter does so **implicitly** (refer to Appendix B). For tackling the CIL problem, we extended both forms of ridge regression to iterative solutions by Theorem 1 and Theorem 2, respectively. We summarize the difference of two methods as follows.
>
> - **Primal Ridge Regression**:
>     - **Advantages**:
>         - Since the dimensionality $d$ is fixed, the computational complexity, mainly determined by matrix inversion, does not increase as data accumulates.
>     - **Disadvantages**:
>         - Requires explicit specification of an appropriate projection function,  which can be challenging to design optimally.
>         - A sufficiently large $d$ is necessary to ensure that the features are expressive enough for accurate classification, which can be computationally expensive.
> - **Dual Ridge Regression**:
>     - **Advantages**:
>         - Leveraging the advantage of the kernel trick, the projection function can be implicitly defined based on the choice of the kernel function, avoiding direct computation and storage of high-dimensional features. It allows for the selection of appropriate kernel functions based on different tasks, enhancing the adaptability of the method.
>         - The method is computationally efficient when the amount of data is small, especially in the few-shot case.
>     - **Disadvantages**:
>         - As the amount of data increases, the computational complexity of kernel matrix ($N \times c$) inversion becomes significant, potentially making the method computational expensive for very large incoming dataset.
> ***
> >***W2** Experiemnts seems a little bit thin in the manucript. Perhaps try to move some of the experiments in the appendix to the main content.*
>
> **A2** We agree with your suggestion to enrich the Experiment section of the main manuscript. We will move a portion of the experiments in the Appendix into the main content for the final version.
> ***
> Based on these additional clarifications, we hope you can keep your support of this work.

---

> ### Comment · Reviewer_rqEC · 2024-08-11
>
> I thank the authors for providing the rebuttal. After reading the rebuttal and other reviewers' comments, all my previous concerns have been adequately addressed. I will keep my positive rating.

---

> > ### Author Response · Authors · 2024-08-11
> > **Thank you for the response**
> >
> > Thank you for reading the response and your support of our work! We are glad that we have addressed all your concerns.

---

### Official Review · Reviewer_Tz45 · 2024-07-15

**Soundness:** 2
**Presentation:** 2
**Contribution:** 3
**Rating:** 6
**Confidence:** 3

**Summary:**

Continual Learning (CL) with Vision-Language Models (VLMs) has a challenge that the model must not forget both previously learned knowledge and VLM pre-trained knowledge. Existing methods realize this by using large-scale reference data or domain identity hints, which is not practical. This paper proposed RAIL, which uses a recursive ridge-regression-based adapter to address the previously seen knowledge while using the zero-shot ability of VLMs for unseen classes. In addition, This paper proposes a novel task X-TAIL that evaluates the model on seen and unseen domains without any domain-identifier hints. The proposed method is empirically evaluated on MTIL and X-TAIL and shows state-of-the-art performance.

**Strengths:**

+ This paper tackled a novel and practical problem, X-TAIL. It is certainly unrealistic to assume that we always have domain information, and it is more natural to consider all classes.
+ The proposed method seemed to have some technical novelty. It is worth commending that the proposed method is not merely an application of adapter methods such as CLIP-Adapter and Tip-Adapter, but has a recursive structure that makes it the method with affinity for CL.
+ The proposed method showed higher performance on both MTIL and X-TAIL than existing methods.

**Weaknesses:**

+ I feel some parts of the paper were difficult to understand.
  + The explanation of the proposed method in the introduction was so unclear that I could not understand the method at all from the introduction section. The author should modify the manuscripts to prepare some belief figures to support the reader’s understanding.
  + I couldn’t understand how to identify ID and OOD classes in the proposed method until I read the appendix. The main manuscript should be self-contained as we can understand it without reading the appendix.
+ The proposed method was evaluated on the few-shot setting in the MTIL evaluation, but it was not evaluated on the normal MTIL.
  + The proposed method should be evaluated on the full data setting following the previous works[9, 14].
+ Since the proposed method is based on ridge regression, there is concern about computational complexity when the amount of data increases. It is necessary to compare and discuss with existing methods, taking into account not only the recognition performance but also the computational efficiency.
+ Typos:
  + eq(1) ||Y-X_e||^2_F -> ||Y-X_eW||^2_F

**Questions:**

+ Why the average Last score of Dual-RAIL is higher than that of fine-tune? Are fine-tune results not the performance in the so-called ideal situation?
+ How are the existing methods evaluated? How were the methods before the advent of CLIP (e.g. LwF, iCaRL) applied for CLIP? How were the MTIL methods applied for X-TAIL?

**Limitations:**

+ As I mentioned in Weaknesses, the proposed method based on ridge regression requires high computational complexity, so it cannot be applicable when the amount of data is very large.

---

> ### Author Rebuttal · Authors · 2024-08-05
>
> Thank you for detailed feedback and valuable suggestions.
> ***
> >***W1** Concerns of paper understanding.*
>
> **A1** Thank you for bringing up your confusions. Based on your suggestion, we decide to make following revisions in the final version.
> - We will update the introduction with clearer explanation of both primal and dual methods. We will include a detailed description of how the CLIP extracted features are projected into higher dimensional space and then classified with ridge regression.
> - We agree that Algorithm 2 in the Appendix provides a clearer explanation of the process for identifying ID and OOD classes. We will move this algorithm to the main manuscript to improve paper's self-containment.
> ***
> >***W2** Evaluation on full data MTIL.*
>
> **A2** We have added experiments of comparing the proposed method with SOTA methods ZSCL[1] and Moe-Adapter[2] under the **full data MTIL setting**, reported **Experiment 1 in the general response.** Our methods consistently surpass existing methods under this setting. We will include the experiments in the final version.
> ***
> >***W3&Limitation** Concerns for large data.*
>
> **A3** Thank you for pointing out the concerns regarding the computational complexity of our method. Our method requires **only forward pass** and matrix computations for parameter updates, **without back-propagation or gradient updates**. Since the backward pass typically accounts for approximately 70% of the entire foward-backward training time[3], and our method trains on every dataset over **only one epoch**, it is faster than back-propagation-based methods. Regarding the impact of the amount of data, we provide the following analysis:
>
> **Primal Ridge Regression**: As detailed in Theorem 1 of our manuscript, the computational complexity for updating parameters mainly relies on the matrix inversion operation in Eqn. 6. Assuming $N$ is the amount of training data and $d$ is the dimension of features, the size of $\mathbf{Φ}^{(n)}$ is $N \times d$, and the size of $\mathbf{M}_p^{(n)}$ is $d \times d$. The large size of the $\mathbf{Φ}^{(n)}\mathbf{M}_p^{(n−1)}\mathbf{Φ}^{(n)T}$ matrix, which is affected by data amount, can be managed by using mini-batches. By breaking down the $\mathbf{Φ}^{(n)}$ into manageable batches and updating $\mathbf{M}_p^{(n)}$ iteratively, our primal method prevents computational complexity from escalating with large data amount.
>
> **Dual Ridge Regression**: We acknowledge that the limitation of high computational complexity in large data scenarios is inherent to kernel-based methods (*e.g.*, kernel SVM), including our dual method. As stated in Theorem 2, the complexity mainly lies in inverting the $\mathbf{K}^{(n)} + \lambda \mathbf{I}$ matrix, where the size of $\mathbf{K}^{(n)}$ is $N \times N$, determined by the amount of data involved in constructing the kernel, making it more suited for few-shot settings. However, existing techniques, such as Nyström approximation[4], can mitigate the computational complexity of kernel-based methods under large data scenario, enhancing the scalability of our method.
>
> Furthermore, to justify the computational efficiency of our method, we added experiments on the comparison of computational times between our proposed RAIL methods and the SOTA methods ZSCL[1] and Moe-Adapter[2] in the X-TAIL setting, reported in **Experiment 2 in the general response**. Our methods are significantly faster than the SOTA methods.
> ***
> >***W4** Typos.*
>
> **A4** We thank the reviewer for pointing out the typo and we will fix it in the final version.
> ***
> >***Q1** Concerns for finetune.*
>
> **A5** We chose finetune as a baseline as it represents a classic method for transferring CLIP to downstream tasks. However, it is important to clarify that finetune is **NOT** the upper bound or the ideal situation for our method.
>
> First, our method differs fundamentally from finetune in terms of optimization. We use ridge regression to achieve a closed-form solution for parameter updates, while finetune relies on back-propagation. The trainable parameters in finetune are those of CLIP encoders, while our method freezes the encoders and only trains additional adapter parameters instead. Notably, our method is proved to be an optimal solution to the optimization problem of joint training, as evidenced by Theorems 1 and 2, making our method equivalent to its own continual learning upper bound.
>
> Second, in the few-shot setting, finetuning the whole encoders of CLIP could potentially lead to overfitting. By contrast, our method freezes the encoders and only trains the adapter via ridge regression, which can overcome overfitting with appropriate regularization parameter, mitigating this risk. Thus, it is reasonable for our method to outperform finetune.
> ***
> >***Q2** How are existing methods evaluated?*
>
> **A6** We replicated existing methods with open-source codes from ZSCL[1] and Moe-Adapter[2] for evaluation. For methods before the advent of CLIP, we adopted implementations from ZSCL's open-source repository. For instance, applying iCaRL to CLIP involves maintaining an exemplar set and updating representations, consistent with the original iCaRL paper. In the X-TAIL setting, we eliminated the specification of domain identity for all methods to ensure a fair comparison between baselines and our method.
> ***
> Given these clarifications, would you consider raising your score for our paper?
> ***
> ***Reference***
>
> [1] Zheng, Zangwei, et al. "Preventing zero-shot transfer degradation in continual learning of vision-language models." *ICCV*. 2023.
>
> [2] Yu, Jiazuo, et al. "Boosting continual learning of vision-language models via mixture-of-experts adapters." *CVPR*. 2024.
>
> [3] Huo, Z., Gu, B. and Huang, H. "Decoupled parallel backpropagation with convergence guarantee." *ICML*. PMLR, 2018.
>
> [4] Chen, Y. and Yang, Y.. "Fast statistical leverage score approximation in kernel ridge regression." *International Conference on Artificial Intelligence and Statistics*. PMLR, 2021.

---

> > ### Comment · Reviewer_Tz45 · 2024-08-10
> >
> > Thank you for the response! All my concerns are addressed by the rebuttal comments. In particular, the authors have completely dispel my concerns for the large data, which makes my evaluation to this paper more positive. Thus, I will raise my rating to 6.

---

> > > ### Author Response · Authors · 2024-08-10
> > > **Thank you for the response**
> > >
> > > Thank you so much for your time and support!  We are glad to hear that the response addressed your concerns.

---

### Official Review · Reviewer_ThWm · 2024-07-15

**Soundness:** 3
**Presentation:** 3
**Contribution:** 3
**Rating:** 6
**Confidence:** 3

**Summary:**

This paper proposes a novel setting called Cross-domain Task-Agnostic Incremental Learning (X-TAIL), in which the model is required to incrementally learn from multiple domains and test images from both seen and unseen domains without any domain identity. Additionally, the authors introduce two Regression-based Analytic Incremental Learning (RAIL) methods (primal form and dual form) and validate the effectiveness of these methods both theoretically and experimentally.

**Strengths:**

1. This paper is well-written and includes clear and accurate figures and equations.
2. The method begins with an introduction to primal and dual form ridge regression, analyzing whether these non-linear projections enhance the separability of CLIP features in images from different domains. This analysis motivates the design of the RAIL-adapter, making the overall approach easier to understand.
3. It is intriguing to explore the regression-based analytic IL, leveraging non-linear projection functions from both primal and dual perspectives to enhance the expressiveness of features extracted by the pre-trained CLIP.

**Weaknesses:**

1. There is little difference in performance between the primal and dual ridge regression methods on most datasets, as shown in Figure 6. Could you provide more analysis on this?
2. Could you compare the parameters of primal and dual ridge regression methods? What are the advantages and disadvantages of each?
3. Please provide more explanation of Figure 2. For example, what do the different colored blocks represent?

**Questions:**

See Weakness

**Limitations:**

Yes

---

> ### Author Rebuttal · Authors · 2024-08-04
>
> Thank you for your overall supportive review of our work. Moreover, we are glad that you found this paper is well-written with good soundness and contribution to the NeurIPS community. Please see our responses addressing your specific concerns below:
> ***
> >***W1** There is little difference in performance between the primal and dual ridge regression methods on most datasets, as shown in Figure 6. Could you provide more analysis on this?*
>
> **A1** Thank you for pointing out that the differences in performance between the primal and dual forms of ridge regression might appear minimal in Figure 6. This issue primarily stems from the **scale of the axes** used in the plot, which does not sufficiently emphasize small variations in performance metrics. We understand that this could potentially obscure meaningful distinctions in the comparative evaluation. To provide a clearer picture of the performance differences, **we kindly refer you to Table 1, where the last performance of both methods are detailed.** The results are also copied below. Here, you can observe that the dual method outperforms the primal method by 3.3% on average in the last performance.
>
> |             | Aircraft | Caltech101 | DTD  | EuroSAT | Flowers | Food101 | MNIST | Pets  | Cars  | Sun397 | Average |
> |-------------|:--------:|:----------:|:----:|:-------:|:-------:|:-------:|:-----:|:-----:|:-----:|:------:|:-------:|
> | Primal-RAIL |   41.7   |     94.0   | 66.0 |   86.4  |   **97.2**  |   82.4  |  **93.1** |  83.6 |  75.0 |  71.3  |   79.1  |
> | Dual-RAIL   |   **45.3**   |     **94.2**   | **69.0** |   **87.0**  |   **97.2**  |   87.2  |  93.0 |  **92.4** |  **82.5** |  **76.3**  |   **82.4**  |
>
> ***
> >***W2** Could you compare the parameters of primal and dual ridge regression methods? What are the advantages and disadvantages of each?*
>
> **A2 Parameter Differences**:
> - **Primal Form**: The parameter matrix **W** in primal ridge regression is determined by the dimensionality $d$ of the projected features, which is a $d \times c$ matrix where $c$ represents the number of classes. The dimension $d$ remains fixed, meaning that the complexity associated with the solution of parameter matrix **W** does not increase as the amount of data grows.
> - **Dual Form**: The parameter matrix **α** in dual ridge regression is a $N \times c$ matrix, where $N$ is the number of data points involved in constructing the kernel matrix. As training progresses and more data is incorporated, *i.e.*, $N$ increases, the computational demands of the method get impacted.
>
> **Advantages and Disadvantages**:
> - **Primal Ridge Regression**:
>     - **Advantages**:
>         - Since the dimensionality of features $d$ is fixed, the computational complexity, mainly determined by matrix inversion, does not increase as data accumulates.
>     - **Disadvantages**:
>         - Requires explicit specification of an appropriate projection function,  which can be challenging to design optimally.
>         - A sufficiently large $d$ is necessary to ensure that the features are expressive enough for accurate classification, which can be computationally expensive.
> - **Dual Ridge Regression**:
>     - **Advantages**:
>         - Leveraging the advantage of the kernel trick, the projection function can be implicitly defined based on the choice of the kernel function, avoiding direct computation and storage of high-dimensional features. It allows for the selection of appropriate kernel functions based on different tasks, enhancing the adaptability of the method.
>         - The method is computationally efficient when the amount of data is small, especially in the few-shot case.
>     - **Disadvantages**:
>         - As the amount of data increases, the computational complexity of kernel matrix ($N \times c$) inversion becomes significant, potentially making the method computational expensive for very large incoming dataset.
>
> ***
> >***W3** Please provide more explanation of Figure 2. For example, what do the different colored blocks represent?*
>
> **A3** Blocks in Figure 2 represents the classification performance of the model across all domains after learning each specific domain.
> - The **blue blocks** in the upper-right matrix indicate the model's zero-shot performance on these domains before learning these domains. These blocks are utilized to evaluate the ability of continual learning methods to *“transfer”* zero-shot capabilities of the VLM.
> - The **gray and green blocks** under the diagonal show the classification performance on these domains after the model has learned these domains. Specifically, the **green blocks** represent the model's “*last”* performance on these domains after learning all domains. These blocks evaluate the adaptability of continual learning methods to new domains and their ability to retain the newly acquired knowledge throughout the continual learning process.
> - The **orange blocks** indicate the *“average”* performance across all time stamps for each domain.
> ***
> We will update the explanations mentioned above into the caption of Figure 2 and the corresponding subsection to clarify the metrics and facilitate easier understanding. In light of these clarifications, would you consider increasing your score for our paper? Otherwise, could you let us know any additional changes you would like to see in order for this work to be accepted?

---

> > ### Comment · Reviewer_ThWm · 2024-08-09
> >
> > Thank you for the response. It has addressed all my concerns. After also considering the comments from the other reviewers, I will raise my rating to 6.

---

> > > ### Author Response · Authors · 2024-08-09
> > > **Thank you for the response**
> > >
> > > Thank you for taking the time to read our response and increasing your score! We are glad to hear that the response addressed your concerns.

---

### Author Rebuttal · Authors · 2024-08-05

We appreciate for all reviewers' insightful and valuable comments. We thank Reviewer ThWm and Reviewer rqEC, who agree that our work is **well-written** and **easy to follow**. We are pleased that Reviewer Tz45 finds **our setting more realistic**, recognizing our efforts to address the requirement for domain information in the existing setting. We are grateful for the recognition from Reviewer ThWm, Reviewer Tz45 and Reviewer rqEC of the **novelty** of our approach and its effectiveness in **addressing the problem of forgetting in continual learning through a recursive solution**.

We have provided detailed, point-by-point responses to address all comments and concerns raised by the reviewers. Additionally, we have conducted the following experiments to further validate our method and address the issues raised.

**1. Full-Shot MTIL Setting**. Following the previous works[1,2], we additionally evaluated our methods on full data MTIL setting. We compared our methods with SOTA methods, ZSCL[1] and MoE-Adapter[2].

||Aircraft|Caltech101|Cifar100|DTD|EuroSAT|Flowers|Food101|Mnist|Pets|Cars|Sun397|***Average***|
|:---:|:---:|:---:|:---:|:---:|:---:|:---:|:---:|:---:|:---:|:---:|:---:|:---:|
|**Transfer**||||||||||||
|ZSCL|--|86.0|67.4|**45.4**|50.4|69.1|87.6|**61.8**|86.8|60.1|**66.8**|68.1|
|MoE-Adapter|--|87.9|**68.2**|44.4|49.9|70.7|**88.7**|59.7|64.5|64.5|65.5|68.9|
|Primal-RAIL|--|**88.4**|**68.2**|44.6|**54.9**|**71.0**|88.5|59.6|**89.0**|**64.7**|65.2|**69.4**|
|Dual-RAIL|--|**88.4**|**68.2**|44.6|**54.9**|**71.0**|88.5|59.6|**89.0**|**64.7**|65.2|**69.4**|
|**Average**||||||||||||
|ZSCL|45.1|92.0|80.1|64.3|79.5|81.6|**89.6**|**75.2**|88.9|64.7|**68.0**|75.4|
|MoE-Adapter|50.2|91.9|**83.1**|69.4|78.9|84.0|89.1|73.7|89.3|67.7|66.9|76.7|
|Primal-RAIL|51.9|95.8|80.1|70.3|81.1|86.1|89.0|73.9|**90.2**|68.4|66.4|77.6|
|Dual-RAIL|**52.5**|**96.0**|80.6|**70.4**|**81.3**|**86.3**|89.1|73.9|**90.2**|**68.5**|66.5|**77.8**|
|**Last**||||||||||||
|ZSCL|40.6|92.2|81.3|70.5|94.8|90.5|**91.9**|98.7|**93.9**|85.3|**80.2**|83.6|
|MoE-Adapter|49.8|92.2|**86.1**|78.1|95.7|94.3|89.5|98.1|89.9|81.6|80.0|85.0|
|Primal-RAIL|51.9|96.5|82.8|80.0|96.0|98.7|89.7|**98.8**|93.3|84.8|78.7|86.5|
|Dual-RAIL|**52.5**|**96.8**|83.3|**80.1**|**96.4**|**99.0**|89.9|**98.8**|93.5|**85.5**|79.2|**86.8**|
***
**2. Computational Efficiency**. We added experiments on the computational efficiency by comparing the computational times of our proposed RAIL methods with SOTA methods, ZSCL[1] and MoE-Adapter[2], in the X-TAIL setting. Our methods are significantly **faster** due to their **one-epoch** nature. (Hardware: i9-13900K & RTX 4090)

||Real time|
|:-------------:|:--------:|
|ZSCL| 514m40.163s|
|Moe-Adapter|47m2.319s|
| Primal-RAIL|**4m0.071s** |
| Dual-RAIL|4m13.200s|
***
**3. Inclusion of CIFAR-100 in X-TAIL setting**. We compared our methods with SOTA methods, ZSCL[1] and MoE-Adapter[2].

||Aircraft |Caltech101|Cifar100|DTD| EuroSAT | Flowers | Food101|Mnist | Pets | Cars | Sun397 |***Average***|
|:---:|:---:|:---:|:---:|:---:|:---:|:---:|:---:|:---:|:---:|:---:|:---:|:---:|
|**Transfer**|||||||||||||
|ZSCL|--|63.7|37.2|32.1|15.8|60.1|82.9|32.1|82.6| 53.3|53.6|51.3|
|MoE-Adapter|--|64.2|35.9| 32.9|17.3|60.6|**86.6**|23.0|**87.2**|63.7|57.1|52.9|
|Primal-RAIL|--|**69.7**|**37.3**|**36.5**|**36.6**|**60.7**|84.0|**46.6**| 86.7 |**66.1**|**62.5**|**58.7**|
|Dual-RAIL|--|**69.7**|**37.3**|**36.5**|**36.6**|**60.7**|84.0|**46.6**| 86.7 |**66.1**|**62.5**|**58.7**|
|**Average**|||||||||||||
|ZSCL|33.4|57.9|41.0|37.7|20.3|68.1|84.0|36.1|82.0|57.7|55.2|52.1|
|MoE-Adapter|42.4|66.4|55.3|49.0|38.3|74.9|**86.2**|46.7|87.4|66.2|58.4|61.0|
|Primal-RAIL|42.4|88.5 |57.1|55.7|64.7|80.7|83.0|62.9|84.8|68.7|63.7|68.4|
|Dual-RAIL|**45.0**|**88.8**|**57.8**|**56.8**|**66.2**|**81.0**|85.2|**63.4**|**87.8**|**68.9**|**64.7**|**69.6**|
|**Last**|||||||||||||
|ZSCL|31.4|59.6|43.9|39.7|28.4|71.6|86.4|40.7|82.6|77.0|70.8|57.5|
|MoE-Adapter|41.8|66.2|59.5|53.7|45.9|84.3|85.8|86.8|87.7|76.2|71.7|69.1|
|Primal-RAIL|41.9|94.0|73.7|67.8|84.4|97.0|83.4|92.6|86.9|75.7|71.4|79.0|
|Dual-RAIL|**45.2**|**94.4**|**74.7**|**70.7**|**87.3**|**97.9**|**86.5**|**92.8**|**91.9**|**81.7**|**76.7**|**81.8**|
***
**4. X-TAIL setting with another order**. We evaluated our methods in the X-TAIL setting with the same order as MTIL order 2, consistent with previous works[1,2]. Our methods are compared with SOTA methods, ZSCL[1] and MoE-Adapter[2].

||Cars|Food101|Mnist|Pets|Flowers|Sun397|Aircraft|Caltech101|DTD|EuroSAT|***Average***|
|:---:|:---:|:---:|:---:|:---:|:---:|:---:|:---:|:---:|:---:|:---:|:---:|
|**Transfer**|||||||||||||
|ZSCL|--|**87.7**|37.3|84.0|**63.6**|63.1|19.7|71.1|34.3|**39.5**|55.6|
|MoE-Adapter|--|83.7|20.6|87.2|62.2|58.4|18.0|70.1|35.6|19.9|50.6|
|Primal-RAIL|--|84.0| **46.7** | **86.7** | 63.5 | **63.7** | **23.5** | **76.8** | **37.3** | 36.7 | **57.7**|
|Dual-RAIL|--|84.0| **46.7** | **86.7** | 63.5 | **63.7** | **23.5** | **76.8** | **37.3** | 36.7 | **57.7** |
|**Average**|||||||||||||
|ZSCL|75.2|**88.9**|48.6|86.8|77.6|67.9|27.7|72.7|38.5|**42.3**|62.6|
|MoE-Adapter|75.4|85.3|64.0|87.4|80.2|65.6|27.6|72.6|40.9|22.2|62.1|
|Primal-RAIL | 78.0 | 83.2 | 65.1 | 88.4 | 83.6 | 67.5 | 31.0 | 80.9 | 43.1 | 41.7 | 66.3 |
|Dual-RAIL | **81.8** | 85.8 | **65.5** | **89.5** | **83.9** |**69.5** | **31.6**| **81.0** |  **43.9** | 41.8| **67.4** |
|**Last**|||||||||||||
|ZSCL|71.9|**88.8**|50.6|88.9|87.2|71.1|38.6|76.1|55.6|67.7|69.7|
|MoE-Adapter|75.1|86.0|79.2|87.4|88.7|72.7|42.2|78.4|61.6|51.3|72.3|
|Primal-RAIL|77.8|83.3|91.4|86.3|97.0|71.6|42.1|94.8|66.3|86.9|79.8|
|Dual-RAIL|**81.8**|86.3|**92.8**|**91.3**|**97.6**|**75.6**|**45.3**|**95.1**|**70.7**|**87.5**|**82.4**|
***
***Reference***

[1] Zheng, Zangwei, et al. "Preventing zero-shot transfer degradation in continual learning of vision-language models." *ICCV*. 2023.

[2] Yu, Jiazuo, et al. "Boosting continual learning of vision-language models via mixture-of-experts adapters." *CVPR*. 2024.

---

### Decision · Program_Chairs · 2024-09-25

**Decision:**

Accept (poster)

**Comment:**

Reviewers agree on acceptance based on the practical problem setting, technical novelty, and state-of-the-art performance. The authors are encouraged to include discussions and results in the rebuttal to the final version.